# A framework for exhaustively mapping functional missense variants

Jochen Weile[1,2,3,4,†] iD, Song Sun[1,2,3,4,5,†], Atina G Cote[1,2,3], Jennifer Knapp[1,2,3], Marta Verby[1,2,3], Joseph C Mellor[2,6], Yingzhou Wu[1,2,3,4], Carles Pons[7], Cassandra Wong[1,2], Natascha van Lieshout[1], Fan Yang[1,2,3,4], Murat Tasan[1,2,3,4], Guihong Tan[2,3], Shan Yang[8], Douglas M Fowler[9], Robert Nussbaum[8], Jesse D Bloom[10], Marc Vidal[11,12] iD, David E Hill[11] iD, Patrick Aloy[7,13] & Frederick P Roth[1,2,3,4,14,*] iD

## Abstract

Although we now routinely sequence human genomes, we can confidently identify only a fraction of the sequence variants that have a functional impact. Here, we developed a deep mutational scanning framework that produces exhaustive maps for human missense variants by combining random codon mutagenesis and multiplexed functional variation assays with computational imputation and refinement. We applied this framework to four proteins corresponding to six human genes: UBE2I (encoding SUMO E2 conjugase), SUMO1 (small ubiquitin-like modifier), TPK1 (thiamin pyrophosphokinase), and CALM1/2/3 (three genes encoding the protein calmodulin). The resulting maps recapitulate known protein features and confidently identify pathogenic variation. Assays potentially amenable to deep mutational scanning are already available for 57% of human disease genes, suggesting that DMS could ultimately map functional variation for all human disease genes.

**Keywords** complementation; deep mutational scanning; genotype–phenotype; variants of uncertain significance

**Subject Categories** Chromatin, Epigenetics, Genomics & Functional Genomics; Genome-Scale & Integrative Biology; Methods & Resources

**Mol Syst Biol. (2017) 13: 957**

## Introduction

Millions of people will soon have their genomes sequenced. Unfortunately, we have only a limited ability to interpret personal genomes, each carrying 100–400 rare missense variants (The 1000 Genomes Project Consortium, 2015) of which many must currently be classified as "variants of uncertain significance" (VUS). For example, gene panel sequencing aimed at identifying germline cancer risk variants in families yielded VUS for the majority of missense variants (Maxwell et al, 2016). Functional variants can be predicted, but when high precision is required, computational tools (Adzhubei et al, 2010; Choi et al, 2012) detect only one-third as many pathogenic variants as experimental assays (Sun et al, 2016). Unfortunately, validated experimental assays enabling rapid clinical interpretation of variants are not available for the vast majority of human disease genes.

Deep mutational scanning (DMS) (Fowler et al, 2010; Fowler & Fields, 2014; Starita et al, 2017), a strategy for large-scale functional testing of variants, can functionally annotate a large fraction of amino acid substitutions for a substantial subset of residue positions. Recent DMS studies, for example, covered the critical RING domain of BRCA1 (Starita et al, 2015) associated with breast cancer risk, and the PPARG protein associated with Mendelian lipodystrophy and increased risk of type 2 diabetes (Majithia et al, 2016). Such maps can accurately identify functionality of a clinical variant in advance of that variant's first clinical presentation. Diverse assays can be used for DMS (see Table EV1). Functional complementation

---

1 Lunenfeld-Tanenbaum Research Institute, Mount Sinai Hospital, Toronto, ON, Canada
2 The Donnelly Centre, University of Toronto, Toronto, ON, Canada
3 Department of Molecular Genetics, University of Toronto, Toronto, ON, Canada
4 Department of Computer Science, University of Toronto, Toronto, ON, Canada
5 Department of Medical Biochemistry and Microbiology, Uppsala University, Uppsala, Sweden
6 SeqWell Inc, Boston, MA, USA
7 Institute for Research in Biomedicine (IRB Barcelona), The Barcelona Institute for Science and Technology, Barcelona, Catalonia, Spain
8 Invitae Corp., San Francisco, CA, USA
9 Department of Genome Sciences, University of Washington, Seattle, WA, USA
10 Fred Hutchinson Research Center, Seattle, WA, USA
11 Center for Cancer Systems Biology (CCSB), Dana-Farber Cancer Institute, Boston, MA, USA
12 Department of Genetics, Harvard Medical School, Boston, MA, USA
13 Institució Catalana de Recerca I Estudis Avançats (ICREA), Barcelona, Catalonia, Spain
14 Canadian Institute for Advanced Research, Toronto, ON, Canada
*Corresponding author. Tel: +1 416 946 5130; E-mail: fritz.roth@utoronto.ca
†These authors contributed equally to this work

---

assays test the variant gene's ability to rescue the phenotype caused by reduced activity of the wild-type gene (or its ortholog in the case of trans-species complementation) (Lee & Nurse, 1987; Osborn & Miller, 2007). Cell-based functional complementation assays can accurately identify disease variants across a diverse set of human disease genes (Sun *et al*, 2016).

Challenges to the DMS strategy include the need to establish robust assays measuring each variant's impact on the disease-relevant functions of a gene, and to generate maps that cover all possible amino acid changes. Also, published DMS maps have not typically controlled the overall quality of measurements nor estimated the quality of individual measurements. Thus, the use of DMS maps to confidently evaluate specific variants has been limited.

Here, we describe a modular DMS framework to generate complete, high-fidelity maps of variant function based on functional complementation. This framework combines elements of previous DMS studies, uses machine learning to impute and improve the map with surprisingly high accuracy, and yields a confidence measure for each reported measurement. In the following sections, we give an overview of the overall framework for DMS, describe its initial application to the SUMO E2 conjugase UBE2I, present complete high-fidelity maps for three new disease-associated proteins and explore the potential for clinical relevance. Finally, we assemble information on functional assays for known human disease genes and conclude that DMS is already potentially extensible to the majority of human disease genes, suggesting the possibility of exhaustive maps of functional variation covering all human genes.

## Results

We describe a framework for comprehensively mapping functional missense variation, organized into six stages (see Fig 1A): (i) mutagenesis; (ii) generation of a variant library; (iii) selection of functional variants; (iv) readout of the selection results and analysis to produce an initial sequence-function map; (iv) computational analysis to impute missing values; and (vi) computational analysis to refine measured values via machine learning. We describe and contrast two versions of this framework: DMS-BarSeq and DMS-TileSeq.

### A barcode-based deep mutational scanning strategy

We first describe DMS-BarSeq and its application to map functional missense variation for the SUMO E2 conjugase UBE2I. In DMS-BarSeq, a heterogeneous pool of cells bearing a library of different barcoded expression plasmid is quantified via barcode-sequencing before and after selection. For Stage 1 of the DMS framework—mutagenesis—we sought a relatively even representation of all possible single amino acid substitutions. We wished to allow multiple mutations per clone, both because this allowed for greater mutational coverage for any given library size, and offered an opportunity to discover intragenic epistatic relationships. To this end, we scaled up a previous mutagenesis protocol (Seyfang & Huaqian Jin, 2004) to develop Precision Oligo-Pool based Code Alteration (POPCode), which yields random codon replacements (see Materials and Methods).

For Stage 2 of the framework—generation of a variant library—we employed *en masse* recombinational cloning of mutagenized UBE2I amplicons into a pool of randomly barcoded plasmids (see Materials and Methods). The full-length UBE2I sequence and barcode of each plasmid were established using a novel sequencing method called KiloSeq which combines plate-position-specific index sequences with Illumina sequencing to carry out full-length sequencing for thousands of samples (see Materials and Methods). We retained clones that carried at least one amino acid substitution to generate a final library comprised of 6,553 UBE2I variants, covering different combinations of 1,848 (61% of all possible) unique amino acid changes. Variant plasmids were pooled, together with empty vector and wild-type control plasmids (see Materials and Methods).

For Stage 3—selection for clones encoding a functional protein—we employed a *S. cerevisiae* functional complementation assay (Jiang & Koltin, 1996; Sun *et al*, 2016), based on human *UBE2I*'s ability to rescue growth at an otherwise-lethal temperature in a yeast strain carrying a temperature-sensitive (ts) allele of the *UBE2I* ortholog *UBC9*. Despite a billion years of divergence, yeast functional complementation assays can accurately discriminate pathogenic from non-pathogenic human variants (Sun *et al*, 2016). The plasmid library from Stage 3 was transformed *en masse* into the appropriate ts strain. Pools were grown for 48 h at the permissive (25°C) and selective (37°C) temperatures, respectively (see Materials and Methods).

To assess variant functions (Stage 4), barcodes were sequenced at multiple timepoints of the selection, enabling reconstruction of individual growth curves and normalized fitness quantification for each of the 6,553 barcoded strains. Functional complementation scores were calibrated so that 0 corresponds to the fitness of the null allele and 1 to wild-type complementation (see Materials and Methods). Using replicate agreement and extent of library representation, we estimated our uncertainty in each fitness value (see Materials and Methods).

Before further refinement in Stages 5 and 6, we wished to assess the quality of the DMS-BarSeq complementation scores. Based on both technical (Fig 1B, top) and biological replicates (different clones carrying the same mutation; Fig 1B, bottom), we found scores to be reproducible (Pearson's *R* of 0.97 and 0.78, respectively). Semi-quantitative manual complementation assays for a subset of mutants that spanned the range of fitness scores (see Materials and Methods) correlated well with DMS scores. Indeed, agreement between large-scale and manual scores was on par with agreement between internal replicates of the large-scale scores (Fig 1B and C).

We also examined evolutionary conservation and computational predictors of deleteriousness, such as PolyPhen-2 (Adzhubei *et al*, 2010) and PROVEAN (Choi *et al*, 2012). Although each is an imperfect measure of the functionality of amino acid changes (Sun *et al*, 2016), each should and did correlate with DMS results (Fig 1D top panel, Appendix Fig S1). Finally, we confirmed that, as expected, amino acid residues on the protein surface are more tolerant to mutation than those in the protein core or within interaction interfaces (Fig 1D, bottom panel). Taken together, these observations support the biological relevance of the DMS-BarSeq approach.

### A tiled-region strategy for mapping functional variation

While DMS-BarSeq has several advantages (see Discussion), its performance comes at the cost of producing an arrayed library of

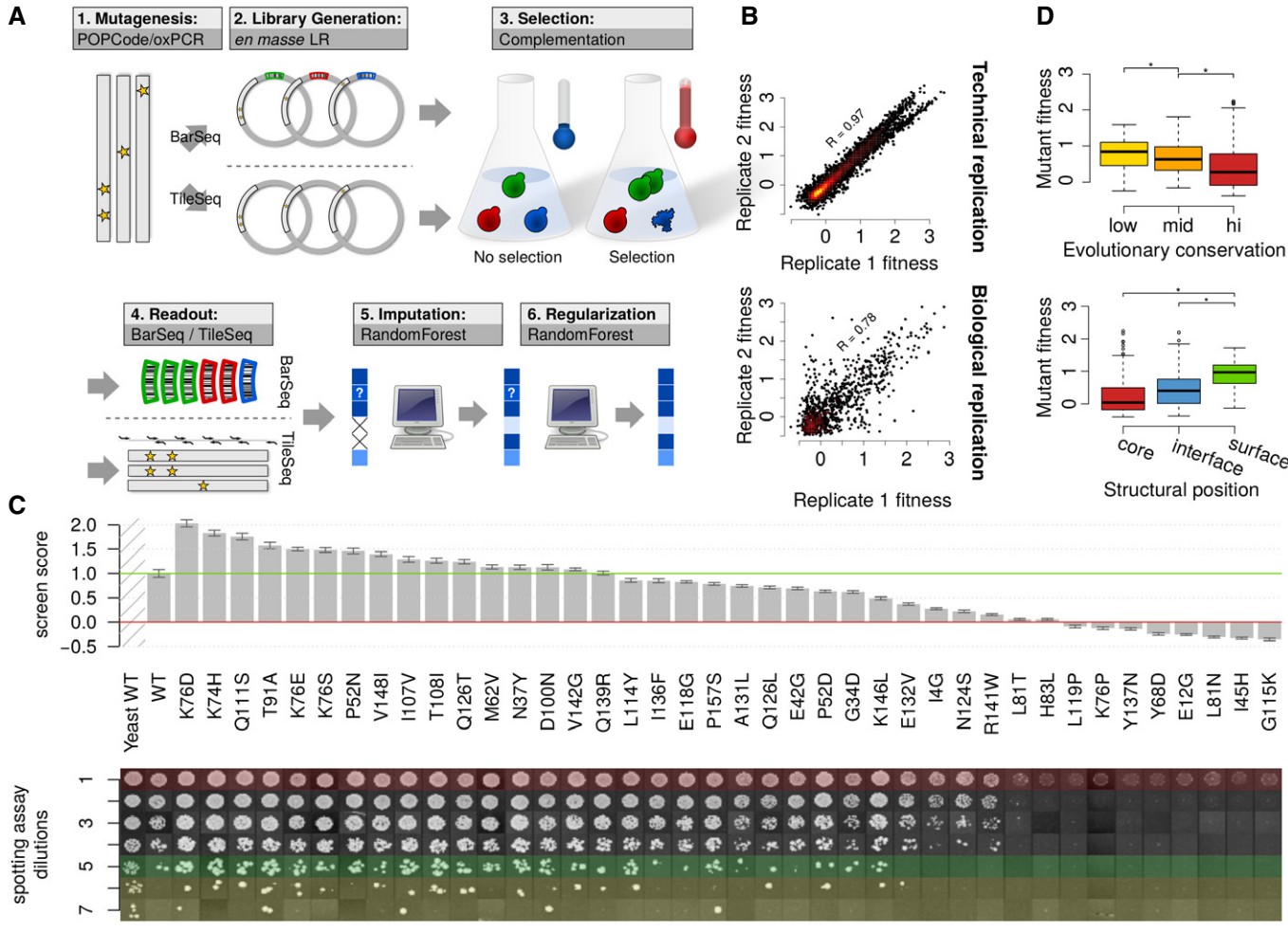

**Figure 1. UBE2I screening and validation.**

A   Modular structure of the screening framework.

B   Raw DMS-BarSeq fitness scores in technical replicates (separately plated assays of the same pool) and biological replicates (separate sub-strains in the pool carrying the same variants).

C   Manual spotting assay validation of a representative set of variants. Each row represents a consecutive fivefold dilution. Marked in red: maximal dilution visible in empty vector control. Marked in green: maximal dilution with visible human wt control. Marked in yellow: dilution steps exceeding visible human wt control. Bar heights represent summary screen scores. Error bars show Bayesian regularized standard error based on three technical replicates and a prior based on pre-selection counts and final score (see Materials and Methods for details).

D   Variants grouped by evolutionary conservation (AMAS score) of their respective sites (top) and grouped by structural context within the protein core, within protein–protein interaction interfaces or on remaining protein surface (bottom). Boxes range across the second and third quartiles with the middle bar representing the median. Whiskers show the most extreme values within 1.5×IQR. As normality cannot be assumed for the distributions of fitness scores, one-sided two-sample Wilcoxon–Mann–Whitney tests were used. Low conservation ($n = 60$ clones) vs. medium conservation ($n = 105$ clones) W = 3789, *$P = 0.015$; medium conservation ($n = 105$ clones) vs. high conservation ($n = 404$ clones) W = 28043, *$P = 1.8 \times 10^{-7}$; Core ($n = 208$ clones) vs. surface ($n = 42$ clones) W = 1649, *$P = 1.01 \times 10^{-10}$; interface ($n = 215$ clones) vs. surface ($n = 42$ clones) W = 2461, *$P = 1.58 \times 10^{-6}$.

clones, each with known coding and barcode sequence. We therefore also evaluated an alternative approach, DMS-TileSeq in which each functional variant is detected via the effect of selection on the abundance of clones carrying that variant. The frequency of each variant in the pool is determined, before and after selection, by deep sequencing of short amplicons that tile the complete coding region.

In terms of mutagenesis (Stage 1), DMS-TileSeq is identical to DMS-BarSeq. Given the mutagenized amplicon library, the cloning step (Stage 2) was carried out by *en masse* recombinational subcloning into expression vectors (thereby skipping the step of arraying and sequencing individual clones). This plasmid pool was

next transformed *en masse* into the *ubc9-ts* strain. Deep sequencing detected 97% of all possible missense variants in our expression library, and 100% of the amino acid substitutions that can be achieved via single-nucleotide mutation. As with DMS-BarSeq, DMS-TileSeq employs pooled strains grown competitively (Stage 3) at the permissive and selective temperatures. In Stage 4, like some previous DMS efforts (Doud & Bloom, 2016), we directly sequenced the coding region from the clone population to determine variant frequency before and after selection. Use of tiled amplicons enables individual template molecules to be sequenced on both strands, allowing elimination of most base-calling errors (Fowler *et al*, 2010;

Whitehead *et al*, 2012; Zhang *et al*, 2016) (see Materials and Methods for details). This reduction in base-calling error allows us to more accurately measure lower allele frequencies in mutagenized libraries.

To further assess the reliability of DMS-TileSeq, we compared results with DMS-BarSeq for UBE2I. DMS-TileSeq and DMS-BarSeq correlation was similar to that observed between biological DMS-BarSeq replicates (Pearson's $R = 0.75$, Appendix Fig S2). DMS-TileSeq and DMS-BarSeq also behaved similarly in their agreement with manual complementation assays (Appendix Fig S3). Thus, DMS-TileSeq avoids the substantial cost of arraying and sequencing thousands of individual clones, while performing on par with DMS-BarSeq in terms of reliability of functional complementation scores.

After using regression to transform the DMS-TileSeq scores to the more intuitive scale of DMS-BarSeq (where 0 corresponds to the median score of null mutant controls and 1 corresponds to the median score of wild-type controls), we combined scores from the two methods, giving greater weight to more confident measurements (see Materials and Methods). Scores emerging from this procedure are referred to as "joint scores" below.

## Machine learning to complete and refine maps

Although nearly all missense variants can be detected in our UBE2I TileSeq libraries, we only considered those variants present with "allele frequency" sufficient to allow confident detection of allele frequency reduction post-selection (see Materials and Methods). After filtering, 2,563 of 3,012 possible amino acid changes (85%) were well measured. To complete missing entries in the map (Stage 5 in the framework), we trained a Random Forest (Breiman, 2001)

regression model using the existing joint scores in the map. The model used four types of predictive feature: intrinsic (derived from other measurements in our map); conservation-based; chemico-physical; and structural. Particularly predictive features (Fig 2D) included the average score of observed substitutions at a given position, as weighted by measurement confidence. Conservation-based features included BLOSUM62 (Henikoff & Henikoff, 1992), SIFT (Ng & Henikoff, 2001) and PROVEAN (Choi *et al*, 2012) scores, and position-specific AMAS (Livingstone & Barton, 1993) conservation. Chemicophysical features included mass and hydrophobicity of the original and wild-type amino acids, and the difference between them. Structural features included solvent accessibility and burial in interaction interfaces. Where DMS-BarSeq scores for multi-mutant clones were available, we also used the confidence-weighted average score of all clones containing a particular substitution, and variant fitness expected from a multiplicative model (St Onge *et al*, 2007) (see Materials and Methods).

We assessed imputation performance using cross-validation. Surprisingly, the error (root-mean-squared deviation or RMSD) of imputed values (0.33) was on par with that of experimentally measured data (Fig 2A). As an additional validation step, we performed manual complementation assays for a set of UBE2I variants that were not present in the machine-learning training dataset and compared the results against imputed values (Fig 2C), again finding strong agreement. Predictions showed the least error in positions with high mutation density and the most error for hypercomplementing variants, that is, those yielding above-WT fitness levels in yeast (Fig 2B). Although hypercomplementation may indicate that a variant is adaptive in yeast, imputation generally predicted these variants to be deleterious, a hypothesis we explore further below.

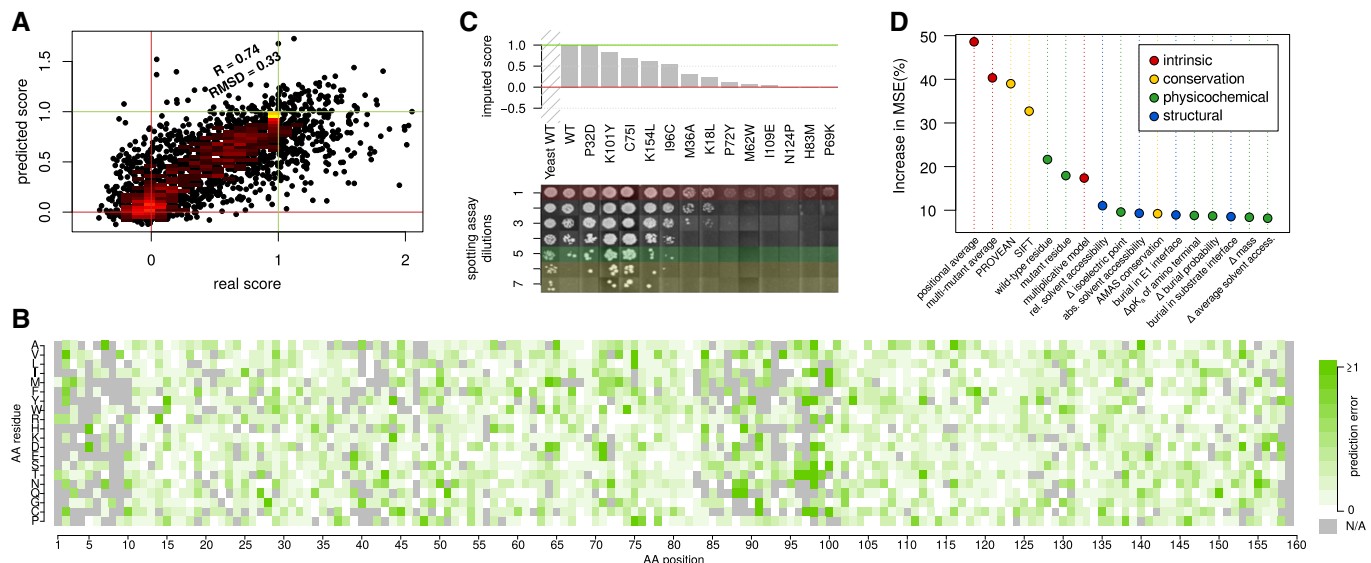

**Figure 2. Validation of machine-learning imputation for UBE2I.**

A  Cross-validation evaluation: Joint scores from DMS-BarSeq and DMS-TileSeq compared to machine-learning prediction in 10× cross-validation. The agreement is comparable to that between biological replicates in the screen itself (compare to Fig 1B).

B  Error map, showing cross-validation results for each data point sorted by amino acid position and mutant residue.

C  Comparison of imputation predictions with individual spotting assays. Each row represents a consecutive fivefold dilution. Marked in red: maximal dilution visible in empty vector control. Marked in green: maximal dilution with visible human wt control. Marked in yellow: dilution steps exceeding visible human wt control.

D  Most informative features in the Random Forest imputation, as measured in % increase in mean squared deviation upon randomization of a given feature.

In order to examine the impact of training set size on the imputation performance, we performed sub-sampling analysis. Performance was poor below ~5% map completeness, increased dramatically at ~10% map completeness, and then improved gradually (and approximately linearly) as completeness levels rose beyond 10% (Appendix Fig S4). We also computationally examined the expected impact of changing the mutagenesis method: When training only on SNP-accessible variants (e.g., if one were using libraries generated by error-prone PCR), imputation RMSD was significantly worse ($P = 2.4 \times 10^{-5}$) compared to a training set of equivalent sample size that can, as POPCode allows, contain all possible amino acid substitutions (Appendix Fig S4).

To refine less-confident experimental measurements (Stage 6 of the framework), we combined joint experimental scores with Random Forest-predicted scores from the imputation procedure, weighting by confidence level. Scores resulting from this combination are referred to as "refined scores" below. Overall, most values were only adjusted minimally through refinement, with 90% of values being altered by < 2.5% of the score difference between null and wt controls (Appendix Fig S5A). This reflects the fact that most values were already of high quality. To evaluate the effect on the minority of variants that required stronger refinement, we looked for cases that were of low quality in the DMS-TileSeq dataset, but well measured in the DMS-BarSeq experiment. These cases would allow us to treat the DMS-BarSeq values as an independent reference for comparison when performing the refinement procedure only on the DMS-TileSeq dataset. We identified six cases that fulfilled these criteria. In all six cases, refinement of DMS-TileSeq resulted in improvement, that is, adjusted the corresponding values such that they more closely resembled the gold standard (Appendix Fig S5B). However, all changes were small, suggesting that our refinement procedure was overly conservative and that alternative weighting schemes should be explored as more "ground truth" data become available.

Manual complementation assays, applied to a set of variants that represented the full range of refined scores (Appendix Fig S3), served to validate the reliability of the complete, refined functional map of UBE2I after imputation and refinement. The map, as seen in Fig 3A, fulfills biochemical expectations, with the hydrophobic core, the active site and protein interaction interfaces being most strongly impacted by mutations (Fig 3B). Detailed observations with respect to structure, biochemistry, and epistatic behavior of double mutants can be found in the Appendix Texts.

## Hypercomplementing variants are likely to be deleterious in humans

We further investigated UBE2I variants exhibiting hypercomplementation (Fig 3A). Manual assays confirmed that complementation with these mutants allows greater yeast growth than does the wild-type human protein (Appendix Fig S6A). These hypercomplementing substitutions did not reliably correspond to "reversion" substitutions that inserted the corresponding *S. cerevisiae* residue (Appendix Fig S6B). Some substitutions could be adaptive by improving compatibility with yeast interaction partners. Indeed, a comparison with co-crystal structure data (Gareau *et al*, 2012) shows that many of the hypercomplementing residues are on the surface proximal to the substrate, with some directly contacting the

substrate's sumoylation motif (Fig 3C). *In vitro* sumoylation assays performed previously for a small number of UBE2I mutants revealed increased sumoylation for some substrates (Bernier-Villamor *et al*, 2002). Comparing our map with these sumoylation assay results, we saw that cases of hypercomplementation were enriched for substrate specificity shift (Appendix Fig S6C). However, other cases of hypercomplementation hinted at different modes of adaptation (see Appendix Text).

To explore whether variants exhibiting hypercomplementation are more likely beneficial or deleterious in a human context, we used a quantitative phylogenetic approach (Bloom, 2014, 2017) to compare three models relating the (refined) complementation scores to evolutionary preference for an amino acid variant: (i) Evolutionary preference is directly proportional to complementation score; (ii) preference has a ceiling at the wild-type complementation score (values > 1 were set to 1); or (iii) preference is set to the reciprocal of complementation score for mutations with greater-than-wild-type scores, corresponding to a deleterious effect of hypercomplementing mutations. We used the phydms software (Bloom, 2017) to test which of these three approaches best described the evolutionary constraint on a set of naturally occurring UBE2I homologs, using recalculated refined scores that excluded conservation features from the imputation and refinement process, to avoid circularity when using natural sequence data to impute or refine scores. The best fit is achieved by treating variants with greater-than-wild-type complementation in yeast as deleterious in humans (Appendix Table S1). We therefore reinterpreted cases of hyperactive complementation in our map as deleterious and repeated the machine-learning training, imputation and refinement procedure. Repeated cross-validation revealed the new imputed values based on the reinterpreted score matrix to be more reliable (i.e., reducing cross-validation RMSD from 0.33 to 0.24).

## Variant impact maps for five additional disease-implicated genes

Having validated the framework, we sought to map functional variation for disease-relevant genes. We applied the higher-throughput TileSeq approach, coupled with yeast complementation, to a diverse set of genes: SUMO1, for which heterozygous null variants are associated with cleft palate (Andreou *et al*, 2007); thiamine pyrophosphokinase 1 (TPK1), associated with vitamin B1 metabolism dysfunction (Mayr *et al*, 2011); and CALM1, CALM2, and CALM3, associated with cardiac arrhythmias (long-QT syndrome (Crotti *et al*, 2013) and catecholaminergic polymorphic ventricular tachycardia (Nyegaard *et al*, 2012)). Because the three calmodulin genes encode the same polypeptide sequence, performing DMS for CALM1 also provided maps for CALM2 and CALM3.

As no corresponding DMS-BarSeq data were available to facilitate TileSeq score re-scaling for these genes, we rescaled scores such that a score of 0 corresponded to the median of nonsense variants while a score of 1 corresponded to the median of synonymous variants. The Random Forest model underlying imputation and refinement was trained anew for each map. (Differences in the stringency of each selection have the potential to introduce non-linear changes in scale that will differ between maps.) Supporting the quality of the resulting four maps, each map showed clear differences in TileSeq-score distributions between likely neutral (synonymous) and likely deleterious (nonsense) variants (Appendix Fig S7).

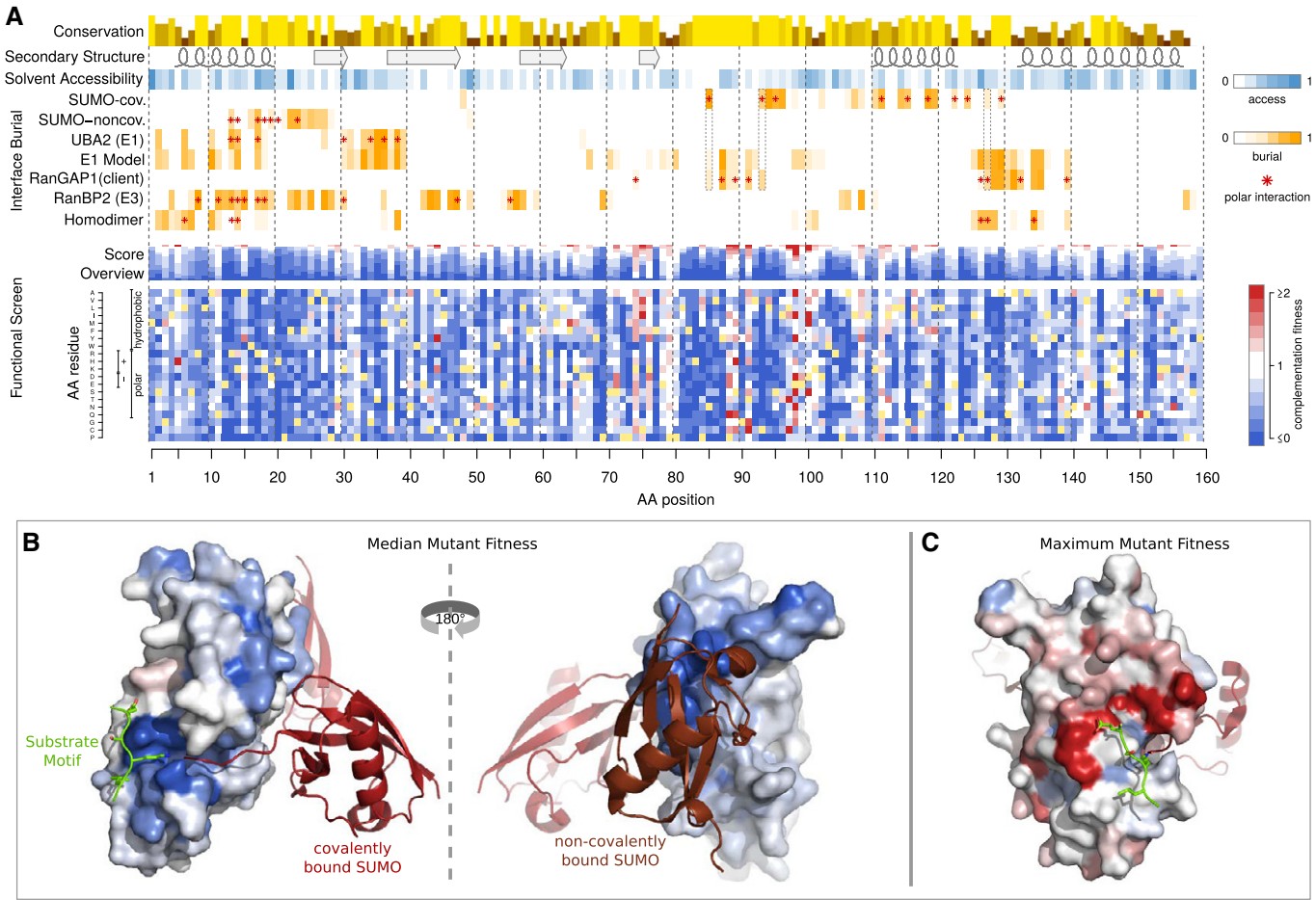

**Figure 3.    A complete functional map of UBE2I.**

A    A complete functional map of UBE2I as resulting from the combination of the complementation screen and machine-learning imputation and refinement. An impact score of 0 (blue) corresponds to a fitness equivalent to the empty vector control. A score of 1 (white) corresponds to a fitness equivalent to the wild-type control. A score > 1 (red) corresponds to fitness above wild-type levels. Shown above, for comparison are sequence conservation, secondary structure, solvent accessibility, and burial of the respective amino acid in protein–protein interaction interfaces with covalently and non-covalently bound SUMO, the E1 UBA2, the sumoylation target RanGAP1, the E3 RanBP2 and UBE2I itself. Hydrogen bonds or salt bridges between residues and the respective interaction partner are marked with red asterisks. Residues buried in both the covalent SUMO and client interfaces are framed with dotted lines, marking the core members of the active site.

B    UBE2I crystal structure with residues colored according to the median mutant fitness. Colors as in (A). The interacting substrate's ΨKxE motif is shown in green stick model; Covalently bound SUMO is shown as a red cartoon model; and non-covalently bound SUMO is shown in brown cartoon model. The structures shown were obtained by alignment of PDB entries 3UIP and 2PE6.

C    UBE2I crystal structure as in (B), with residues colored according to maximum mutant fitness.

To assess the impact of the machine-learning imputation and refinement on the different maps, we measured the completeness of each map before and after imputation, the cross-validation RMSD of the imputation, as well as the maximum standard error value for each map before and after refinement (Table 1). On average, 24.6% of scores were obtained purely by imputation, and 3.96% of scores were appreciably changed by > 5% of the difference between null and wt controls as a result of refinement. Proteins for which overall map quality was initially lower were improved most by refinement, while others, like SUMO1, improved only modestly. Inspection of the maps yielded a number of interesting biochemical and structural observations (see Appendix Texts).

Phylogenetic analysis of SUMO1, as for UBE2I, showed that variants that complement yeast better than wild-type are best modeled as

being deleterious in humans (Appendix Table S1). As was done for the first map, we transformed above-wild-type fitness scores to be deleterious before the imputation and refinement step (see Materials and Methods). Because hypercomplementing substitutions may nonetheless provide interesting clues about differences between yeast and human cellular contexts, we provide both transformed (Fig 4) and untransformed (Appendix Fig S8) map versions. The full numerical values underlying the maps can also be found in Dataset EV1.

**DMS functional maps reflect clinical phenotypes**

To validate the utility of our maps in the context of human disease, we extracted known disease-associated variants from ClinVar (Landrum *et al*, 2016), as well as rare and common polymorphisms

**Table 1.  Map quality comparison.**

| Gene | Possible AA changes | Achieved AA changes | Imputation RMSD | Experimental max(s.e.m.) | Refined max(s.e.m.) | Refinement > 0.05 |
|---|---|---|---|---|---|---|
| UBE2I | 3021 | 2563 (85%) | 0.24 | 0.36 | 0.25 | 2.46% |
| SUMO1 | 1919 | 1700 (89%) | 0.25 | 0.19 | 0.17 | 1.06% |
| TPK1 | 4617 | 3181 (69%) | 0.34 | 0.49 | 0.37 | 5.51% |
| CALM1 | 2831 | 1813 (64%) | 0.29 | 0.28 | 0.22 | 6.84% |

Experimental max(s.e.m.): the largest standard error associated with any experimentally measured score in the given dataset; refined max(s.e.m.): the largest standard error associated with any refined score in the given dataset. Refinement > 0.05: the percentage of variants whose scores were changed by more than 0.05 as a result of refinement.

observed independent of disease from GnomAD (Lek *et al*, 2016), and somatic variants previously observed in tumors from COSMIC (Forbes *et al*, 2001).

While no germline disease-associated missense variants are known for UBE2I and SUMO1 in ClinVar, somatic cancer variants have been observed for both genes according to COSMIC. Somatic variants in these three genes exhibited higher functional impact in DMS maps than germline variants (Wilcoxon $P = 2.6 \times 10^{-5}$) (Fig 5A). This does not necessarily suggest that either of these genes are cancer drivers, as even passenger somatic variants should subject to less purifying selection than germline variants, but it does lend further credence to the biological relevance of our maps.

For TPK1, many very rare variants (minor allele frequency or MAF $< 10^{-6}$) are seen in GnomAD. The majority of these variants were scored as deleterious (Appendix Fig S9A). Thiamine metabolism dysfunction syndrome, reported to be caused by variants in TPK1, is a severe disease to which patients succumb in childhood (Mayr *et al*, 2011). Although GnomAD attempted to exclude subjects with severe pediatric disease, the abundance of rare predicted-deleterious variants may be understood by the disease's recessive inheritance pattern. Using phased sequence data from the 1000 Genomes Project (The 1000 Genomes Project Consortium, 2015) to determine diploid genotypes in TPK1, we assigned each subject a diploid score corresponding to the maximum (refined) score across each pair of alleles. This improved prediction performance markedly, leading to complete separation between disease and non-disease genotypes using DMS, PROVEAN, or PolyPhen-2 scores (Appendix Fig S9B). However, additional compound heterozygotes with known disease status will be required to compare DMS with computational methods in the task of identifying TPK1 disease variants.

Because the inheritance pattern of calmodulin disorders is typically dominant (Crotti *et al*, 2013), we did not consider diploid genotypes but simply evaluated the ability of the (refined) DMS scores to distinguish disease from non-disease variants (Fig 5B). DMS scores performed well according to precision-recall analysis, with an area under the precision-recall curve (AUC) of 0.72, exceeding both PROVEAN (AUC = 0.48) and PolyPhen-2 (AUC = 0.47) (Fig 5C). At a stringent precision threshold of 90%, DMS exceeded twice the sensitivity of PROVEAN and PolyPhen-2.

We further wished to explore how classification based on these observations would perform on variants of uncertain significance (VUS). We therefore examined missense VUS substitutions seen by Invitae, a clinical genetic testing company. Ten rare calmodulin variants had been encountered, of which half were from tests ordered due to a cancer indication, and the other half from tests ordered for a cardiac disease indication. Blinded to indication, we ranked and classified the 10 Invitae VUS variants by DMS score (Table 2). We classified variants as "damaging" if they were below both the highest score of known pathogenic variants and the lowest score of GnomAD variants, and classified variants as "benign" if they were both above the highest-scoring known-pathogenic variant and the lowest-scoring GnomAD variant. All others were classified as "uncertain". Using these criteria, two Invitae variants were classified as damaging, two as uncertain, and six as benign. Based on the patient test indications subsequently revealed by Invitae, five out of the six variants we classified as benign were ordered due to a non-cardiac indication, while both variants with damaging predictions and both with VUS predictions corresponded to cardiac indications. Overall, DMS scores (which do not depend on the somewhat arbitrary classification system described above) showed a significant association with cardiac indications ($P = 0.008$, U = 24.5; Mann–Whitney U-test). We note that the DMS scores used in this analysis (Table 2) differ slightly from those in the final dataset (Dataset EV1) because they were derived from an earlier version of the analysis pipeline. To uphold the integrity of the blinded test, the old values are shown in Table 2, but using the new values yielded precisely the same association between CALM1 DMS score and cardiac indications ($P = 0.008$, U = 24.5; Mann–Whitney U-test).

## Potential for applying deep mutational scanning more widely

DMS mapping requires an *en masse* functional assay that can be applied at the scale of $10^4$–$10^5$ variant clones. Among ~4,000 disease genes, examination of four systematic screens and curated literature suggests that ~5% of human disease genes currently have a yeast complementation assay (Hamza *et al*, 2015; Kachroo *et al*, 2015; Sun *et al*, 2016). This number could grow dramatically via systematic complementation testing under different environments and genetic backgrounds. Moreover, complementation assays can also be carried out in other model systems including human cells (Hart *et al*, 2015). Based on only three large-scale CRISPR studies (Wang *et al*, 2014; Blomen *et al*, 2015; Hart *et al*, 2015), cellular growth phenotypes (which might serve as the basis for an *en masse* selection) have already been observed in at least one cell line for 29% of human disease genes. Beyond complementation, assays of protein interaction can, in addition to identifying variants directly impacting interaction, can detect variants ablating overall function through effects on protein folding or stability. In a recent study, approximately two-thirds of disease-causing variants were found to impact at least one protein interaction (Sahni *et al*, 2015). Although only a

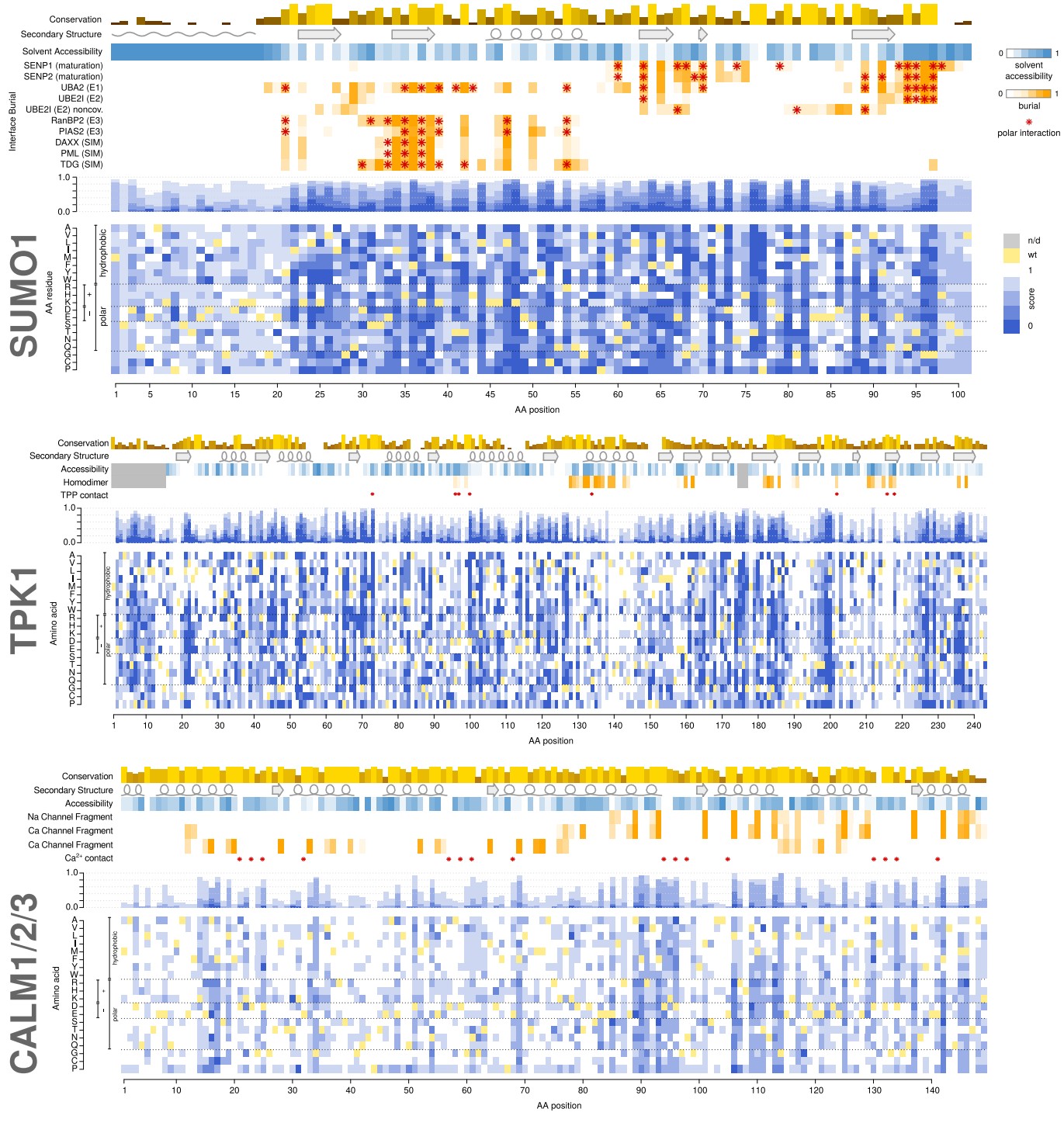

**Figure 4.  Functional maps of SUMO1, TPK1, and calmodulin (CALM1/2/3).**
Layout and colors as in Fig 3.

minority of human protein interactions have been mapped (Rolland *et al*, 2014), already 40% of human genes have at least one interaction partner detectable by yeast two-hybrid assay in a recent screen (Rolland *et al*, 2014). Taking the union of available assays, we estimate that 57% of known disease-associated genes (Dataset EV2) already have an assay that is potentially amenable to DMS.

## Discussion

The framework for systematically mapping functional missense variation we describe here combines elements of previous DMS studies and introduces a new mutagenesis strategy and a machine-learning-based imputation and refinement strategy. This framework

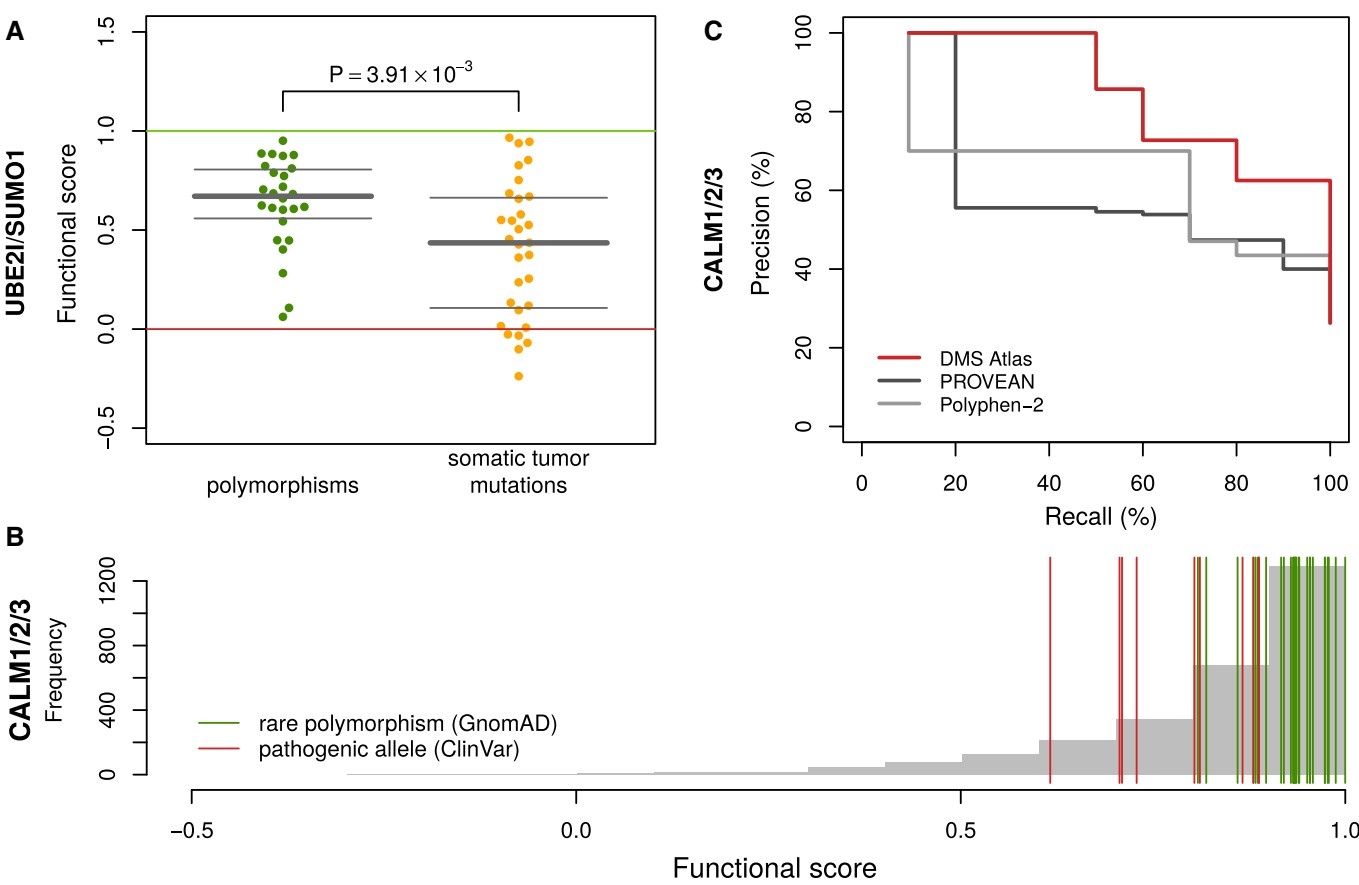

**Figure 5. DMS functional maps reflect clinical phenotypes.**

A Comparison of (refined) functional scores between rare polymorphisms (GnomAD) and somatic tumor mutations (COSMIC) in UBE2I and SUMO1. Bars show median and quartiles. As normality cannot be assumed for the distributions of fitness scores, a one-sided two-sample Wilcoxon–Mann–Whitney test was used: $n = \{26, 31\}$ variants, $W = 570.5$, $P = 3.73 \times 10^{-3}$.

B Impact score distributions in calmodulin overlayed with previously observed alleles in CALM1, CALM2, and CALM3: Rare alleles from GnomAD are shown in green; ClinVar alleles classified as pathogenic are shown in red.

C Precision-recall curves for our DMS atlas, PROVEAN, and PolyPhen-2 with respect to distinguishing Gnomad variants from pathogenic alleles from ClinVar.

enables DMS maps that are "complete" in the sense that high-quality functional impact scores are provided for all missense variants to full-length proteins. Application to four proteins highlighted complex relationships between the biochemical functions of these proteins with phenotypes in the yeast model system. Analysis of pathogenic variation, especially for calmodulin, supported the potential clinical utility of DMS maps from this framework.

The two described versions of DMS, DMS-BarSeq and DMS-TileSeq, each has advantages and limitations (Appendix Table S2). DMS-BarSeq permits study of the combined effects of variants at any distance along the clone and therefore can reveal intramolecular genetic interactions. For DMS-BarSeq, fully sequenced variant clones are arrayed, enabling further investigation of individual variants. DMS-BarSeq can directly compare growth of any clone to null and wild-type controls, resulting in an intuitive scoring scheme. However, despite the efficient KiloSeq strategy for sequencing arrayed clone sets we report for the first time here, DMS-BarSeq is more resource-intensive. Although the regional sequencing strategy of DMS-TileSeq can only analyze fitness of double mutant combinations falling within the same ~150 bp tile, it is far less resource-intensive than DMS-BarSeq.

In the TileSeq libraries, many clones will contain multiple amino acid substitutions (e.g., our UBE2I library averages 2.1 amino acid changes per clone, so in this case, the Poisson distribution predicts that 62% of clones will have more than one amino acid substitution). This raises the concern that the presence of multiple mutations in these clones could obscure the functional effect of any single mutation. However, the DMS-TileSeq scores in our UBE2I map follow a distribution for synonymous variants that is unimodal and distinct from the (also unimodal) distribution for nonsense codon variants, indicating that, despite the presence of multiple variants in many clones, we are able to clearly separate neutral variants from null variants (Appendix Fig S7). Indeed, despite the fact that DMS-TileSeq libraries often have multiple clones, and DMS-BarSeq analysis was based on single-mutant clones, our evaluations of DMS-TileSeq and DMS-BarSeq maps of UBE2I indicated that they are of similar quality. This may be understood by considering the large clone population analyzed (typically > 100,000 clones), which means that the impact of each given query mutation will be an average effect over many genetic backgrounds. This is analogous to detecting a shift in a single SNP's allele frequency between case and

**Table 2.** Invitae VUS classification.

| Variant | MAF | sd/rmsd | Imp/ref | Unrefined | DMS | DMS call | Indication |
|---------|-----|---------|---------|-----------|-----|----------|------------|
| D94A | NA | 0.26 | Imputed | NA | 0.46 | Likely damaging | Cardio |
| D96H | NA | 0.26 | Imputed | NA | 0.72 | Likely damaging | Cardio |
| I28V | $10^{-5}$ | 0.05 | Mild ref. | 0.88 | 0.88 | Uncertain | Cardio |
| N98S | NA | 0.05 | Mild ref. | 0.89 | 0.89 | Uncertain | Cardio |
| T35I | $4 \times 10^{-6}$ | 0.04 | Mild ref. | 0.93 | 0.93 | Likely benign | Non-Cardio |
| E48G | NA | 0.05 | Mild ref. | 0.93 | 0.93 | Likely benign | Cardio |
| G26D | NA | 0.06 | Mild ref. | 0.94 | 0.94 | Likely benign | Non-Cardio |
| T27S | $3 \times 10^{-5}$ | 0.05 | Mild ref. | 0.96 | 0.96 | Likely benign | Non-Cardio |
| V122A | NA | 0.05 | Mild ref. | 0.98 | 0.98 | Likely benign | Non-Cardio |
| A104G | NA | 0.08 | Mild ref. | 1.00 | 1.00 | Likely benign | Non-Cardio |

sd/rmsd, standard error (for measured values)/root-mean-squared deviation (for imputed values); imp/ref, imputation/refinement; mild ref., mild refinement.

control populations in a genomewide association study despite variation at other loci. We cannot exclude the possibility that libraries that have a higher average number of mutations per clone or are from genes for which a higher fraction of missense variants are deleterious, may perform less well in TileSeq. In such cases, it may be desirable to reduce the mutation rate in the POPCode protocol. This can be achieved by using lower concentrations of mutagenic oligos, or using "DMS by parts," in which multiple mutagenized libraries are generated, each focusing mutagenesis on a different segment of the protein.

Given that most missense variants in individual human genes are single-nucleotide variants (Lek *et al*, 2016), and given that only ~30% of all possible amino acid substitutions are accessible by single-nucleotide mutation, one might wonder why codon mutagenesis should be preferred over single-nucleotide mutagenesis. We see four arguments for codon-level mutagenesis: (i) Knowing the functional impact of all 19 possible substitutions at each positions enables clearer understanding of the biochemical properties that are required at each residue position; (ii) an analysis of > 60,000 unphased human exomes (Lek *et al*, 2016) found that each individual human harbors ~23 codons containing multiple nucleotide variants that together could encode an amino acid not encoded by either single variant; (iii) it is not straightforward to generate balanced libraries in which every single-nucleotide variant has roughly equal representation, given that error-prone amplification methods strongly favor transition mutations over transversion mutations, while still avoiding frequent introduction of new stop codons; and (iv) the major cost of DMS will likely continue to be development and validation of the functional assay, so using codon-level mutagenesis instead of (or in addition to) nucleotide-level mutagenesis has a relatively small impact on overall cost.

This study yielded four DMS maps measuring functional impact of ~16,000 missense variants. The maps generated for sumoylation pathway members UBE2I and SUMO1, and disease-implicated genes CALM1/2/3 and TPK1 using our framework were consistent with biochemical expectations while providing new hypotheses. DMS maps based on functional complementation were highly predictive of disease-causing variants, outperforming popular computational prediction methods such as PolyPhen-2 or PROVEAN, confirming previous observations (Sun *et al*, 2016). Given sufficient

experimental data for training, our results show that imputation can "fill the gaps" with scores that are nearly as reliable as experimental measurements and that computational refinement can improve upon experimental measures.

Currently, the machine-learning model underlying our imputation and refinement is re-trained for every new map. Future imputation procedures may benefit by aggregating data from many maps to train a more general imputation model. One challenge will be ensuring that each map is measured on the same scale. For example, the score distributions for missense variants in TPK1 showed a strong bias toward deleteriousness, while missense variants in Calmodulin were biased toward neutrality. However, it is unclear whether this reflects intrinsic properties of these genes as opposed to differences in the stringency of the two selection experiments.

As the community carrying out DMS experiments proceeds toward a (perhaps-distant) common goal of generating a DMS map for all human disease-associated proteins, there will be serious challenges that both TileSeq and our computational methods help to address. Previous DMS maps have assessed variation in relatively short polypeptide regions (typically < 200 amino acids in length). As we approach the median human protein length of ~500 residues, the constraint that we have only ~1–2 missense variants per clones necessarily reduces the mutational density in each clone and thus, the allele frequency in the mutagenized population. This will require a substantial increase in the scale of the library of independently transformed/transfected cells and correspondingly increased sequencing depth to accurately quantify low allele frequencies. Through the reduction in base-calling errors permitted by sequencing both strands, TileSeq allows analysis of lower-allele-frequency variants. Similarly, the need to cover all single amino acid substitutions is substantially ameliorated by our finding that, given a critical mass of DMS data, the quality of imputed scores is nearly as good as experimental measurements. Each of these improvements could have a major impact on the field of deep mutational scanning.

Genome sequencing is likely to become common in clinical practice. Current estimates suggest that every human carries an average of 100–400 rare variants that have never before been seen in the clinic. DMS meets a critical need for fast, reliable interpretation of variant effects. Instead of generating clones and functionally testing

variants of unknown significance after they are first observed, DMS offers exhaustive maps of functional variation that enable interpretation immediately upon clinical presentation, even for rare and personal variation. Our survey of assays revealed that the majority (57%) of human disease genes are potentially already accessible to DMS analysis, so that we may begin to imagine an atlas of DMS maps that reveals pathogenic missense variation for all human disease proteins.

Although our current implementation of the complementation assay uses a temperature-sensitive (ts) allele to provide a control, genes for which no ts allele is yet known are still amenable to DMS by using a null background in combination with an inducibly expressible covering allele. There is even a potential for increasing the number of genes with complementation assays by systematically screening for sensitized backgrounds (exploiting known synthetic lethality relationships or growth conditions). However, growth-based complementation assays have limitations in that they may have limited ability to detect gain- or change-of-function variants. Yeast is also limited as a platform in which to study splicing regulatory or splicing variants. While adaptation of DMS technology to human cell lines will be challenging, recent advances remove some previous hurdles. In addition to the availability of CRISPR-Cas9 to generate homozygous disruptions in target genes, recent advances in "landing pad" technology (Matreyek *et al*, 2017) now allow transfection and integration of a specific sequence into 1–8% of a population of Hek293T cells. Thus, in theory, DMS could be done using on the order of 1 M human cells.

## Materials and Methods

### POPCode mutagenesis

The Precision Oligo-Pool based Code Alteration (POPCode) scales up a previous method (Seyfang & Huaqian Jin, 2004) to achieve coverage over the complete spectrum of possible amino acid changes at all protein positions. POPCode requires design of an oligonucleotide centered on each codon in the open reading frame (ORF) of interest, such that the target codon is replaced with an NNK degenerate codon. This has been previously demonstrated to allow all amino acid changes while reducing the chance in generating stop codons (Pal & Fellouse, 2005). Within each mutagenic oligonucleotide, the arm flanking the target codon is varied to achieve a predicted melting temperature that is as uniform as possible to facilitate an even mutation rate across the ORF sequence. We developed a web tool that automates this design step, available online at http://llama.mshri.on.ca/cgi/popcodeSuite/main (see also "Code and data availability" section).

The POPCode mutagenesis experiment was performed via the following steps: (i) The uracil-containing wild-type template was generated by PCR-amplifying the ORF with dNTP/dUTP mix and Hot Taq DNA polymerase; (ii) the mixture of phosphorylated oligonucleotide pool and uracil-containing template was denatured by heating it to 95°C for 3 min and then cooled down to 4 degrees to allow the oligos hybridize to the template; (iii) gaps between hybridized oligonucleotides were filled with the non-strand-displacing Sulfolobus polymerase IV (*NEB*) and sealed with T4 DNA ligase (*NEB*) and (iv) after degradation of the uracil-doped wild-type

strand using Uracil-DNA glycosylase (UDG) (*NEB*), the mutant strand was amplified with attB-sites-containing primers and subsequently transferred *en masse* to a donor vector by Gateway BP reaction to generate a library of entry clones.

### Synthesis of uracil-containing template

A 50 µl PCR reaction contained the following: 1 ng template DNA, 1× Taq buffer, 0.2 mM dNTPs-dTTP, 0.2 mM dUTP, 0.4 µM forward and reverse oligos, and 1 U Hot Taq polymerase. Thermal cycler conditions are as follows: 98°C for 30 s, 25 cycles of 98°C for 15 s, 60°C for 30 s, and 72°C for 1 min. A final extension was performed at 72°C for 5 min. Uracilated amplicon was gel-purified using the Minelute gel purification kit (Qiagen).

### Phosphorylation of mutagenic oligos

Desalted oligos were purchased from Eurofins or Thermo Scientific. The phosphorylation reaction is as follows: A 50 µl reaction containing 1× PNK buffer, 300 pmoles oligos, 1 mM ATP, and 10 U polynucleotide kinase (NEB) was incubated at 37°C for 2 h. The reaction was used directly in the subsequent POPCode reaction.

### POPCode oligo annealing and fill-in

A 20 µl reaction containing 20 ng uracilated DNA, 0.15 µM phosphorylated oligo pool, and 1.5 µM 5′-oligo was incubated at 95°C for 3 min followed by immediate cooling to 4°C. A 30 µl reaction containing 1× Taq DNA ligase buffer, 0.2 mM dNTPs, 2 U Sulfolobus DNA polymerase IV (NEB), and 40 U Taq DNA ligase (NEB) was added to the DNA and was incubated at 37°C for 2 h.

### Degradation of wild-type template

1 µl fill-in reaction was added to a 20 µl reaction containing 1× UDG buffer and 5 U Uracil-DNA glycosylase (NEB) and incubated at 37°C for 2 h.

Amplification of mutagenized DNA. 1 µl UDG reaction was added to a 50 µl reaction containing 1× Taq buffer, 0.2 mM dNTPs, 0.4 µM forward and reverse oligos, and 1 U Hot Taq polymerase. Thermal cycler conditions are as follows: 98°C for 30 s, 25 cycles of 98°C for 15 s, 60°C for 30 s, and 72°C for 1 min. A final extension was performed at 72°C for 5 min.

### Single-nucleotide mutagenesis

Oxidized nucleotide PCR was performed as previously described by (Mohan *et al*, 2011). Primers were designed to attach attB sites to the product in preparation for Gateway cloning.

### Preparation of oxidized nucleotides

A 100 µM dNTP mixture was incubated at 37°C with 5 mM $FeSO_4$ for 10 min. Addition of 0.5 M mannitol was used to stop the reaction. Oxidized nucleotides were prepared fresh for every PCR reaction.

### PCR in the presence of oxidized nucleotides

PCR reaction containing 1–5 ng template DNA, 1× Thermopol buffer (Invitrogen), 1.5 mM $MgCl_2$, 0.2 mM dNTP, 0.33 µM forward and reverse primers containing attB sites, 1 U Taq polymerase was set up during the nucleotide oxidation reaction. Oxidized nucleotides were the last component added to the PCR reaction at a

concentration of 0.1 mM (half the amount of regular dNTP). Thermal cycler program: 95°C for 10 min, 30 cycles of 95°C for 1 min, 50°C for 1 min, 72°C for 1 min, final extension at 72°C for 10 min. Mutagenized PCR product was visualized on a 1% agarose gel and gel-extracted using a gel extraction kit (Qiagen). The gel-extracted PCR product is the pooled mutagenesis product carrying attB sites that is carried through to the KiloSeq stage.

### Library generation

#### Generation of mutagenized pool of Entries

An en masse Gateway BP reaction containing 150 ng of pooled mutagenesis PCR product carrying attB sites, 150 ng of pDONR223, 1 μl Gateway BP Clonase II Enzyme Mix (*Invitrogen*), 1× TE buffer is prepared. This reaction is incubated overnight at room temperature and then transformed into *E. coli* aiming for the maximum number of transformants (at least 100,000 CFUs) to keep complexity high. Several colonies are picked at this stage for a quality control check by Sanger sequencing, and the rest are put through a pooled DNA extraction. The result is a pool of mutagenized PCR product inserted into the entry vector pDONR223.

#### Generation of barcoded destination pools

Barcoded destination plasmids were generated as previously reported (Yachie *et al*, 2016), but instead of being arrayed were maintained as pools with high complexity. Briefly, a linear PCR product containing two random 25 nucleotide "barcode" regions flanked by loxP and lox2272 sites along with common linker sequences for priming was combined with a gateway compatible vector at a SacI restriction site through *in vitro* DNA assembly (Gibson *et al*, 2009). This barcoded destination vector pool was transformed into One Shot ccdB Survival T1R Competent Cells (Invitrogen). The transformations were spread onto large round LB+ampicillin petri plates for increased selection capacity, and pool complexity was estimated from CFU counts. The plates were combined into a single pool for plasmid DNA extraction by maxiprep.

#### En masse Gateway LR reaction

An en masse Gateway LR reaction was used to transfer the mutagenized pool of entries into the barcoded destination pool. This reaction takes place over 5 days. On Day 1, a 5 μl reaction containing 150 ng of mutagenized ORF pool in pDONR223 backbone, 150 ng barcoded pHYC expression vector pool, 1 μl LR Clonase II Enzyme Mix, 1× TE buffer is prepared. The reaction is incubated at room temperature overnight. Daily, from the second to the fifth day, add in a 5 μl volume consisting of 150 ng barcoded pHYC expression vector, 1 μl LR Clonase II Enzyme Mix, 1× TE buffer, incubating at room temperature overnight each day. On Day 5, the final volume is 25 μl.

#### Transformations and colony picking

LR reactions were transformed into *E. coli* and plated to achieve a density of 400–600 individual colonies per plate. A Biomatrix robot (Biomatrix BM5-BC robot, S&P Robotics) was then used to automatically pick and array 384 colonies per plate for a total of ~20,000 clones in ~52 plates per ORF of interest. Each colony at this stage should contain a pHYC expression vector harboring a variant of the ORF of interest and a unique barcode.

### KiloSeq

For the BarSeq method, to establish the identity of each plasmid barcode and its associated set of mutations in the target ORF we used KiloSeq (Appendix Fig S10) (either carried out in our laboratory or as a service from SeqWell Inc., Beverly, MA, USA). The first step is to PCR-amplify a segment of the plasmid containing both ORF and barcode locus. PCRs were carried out using the Hydrocycler 16 (LGC Group, Ltd.), using primers with well-specific index sequences. Amplicons from each plate were pooled and subjected to Nextera "tagmentation" using Tn5 transposase to generate a library of amplicons with random breaks to which the adapters have been ligated. We then re-amplify those fragments to generate a library of amplicons such that one end of each amplicon bears the well-specific tag and the other "ladder" end bears the Nextera adapter. These libraries can be re-amplified to introduce Illumina TruSeq adaptors, allowing multiple plates of amplicons to be sequenced together. Paired-end sequencing was carried out using Illumina NextSeq 500. In each pair of reads, one read will reveal the well tag and the barcode locus, whereas the other will contain a fragment of the mutant ORF, and these fragments can be assembled into a contiguous sequence.

To perform demultiplexing, barcode identification and insert resequencing, we developed a sequence analysis pipeline (see "Code and data availability" section). In the first step, Illumina bcl2fastq is used to demultiplex the reads at the plate level using the custom Nextera indices. The resulting FASTQ files are then further demultiplexed using the well-tags in a highly parallel fashion. This results in a folder structure containing tens of thousands of individual FASTQ files sorted by plate and well location. These are then further processed in parallel to identify barcodes. Wells can sometimes contain more than one clone (e.g., due to incomplete washing in the robotic pinning process). Thus, barcode sequences are extracted from each read and then clustered by edit distance to determine the set of barcodes in each well. The associated paired reads for each barcodes are then further split by barcode. Each barcode-specific set of ORF reads can then be analyzed with respect to mutations. Bowtie2 software (Langmead & Salzberg, 2012) is used to align reads to the ORF template, PCR duplicates are removed, and nucleotide variants called using samtools pileup (Li *et al*, 2009). Given limited read lengths, identification of longer indels is not straightforward. A solution was found by extracting depth of coverage tracks for each clone and normalizing them with respect to average positional coverage across each 384-well plate, applying an edge-detection algorithm to find sudden increases or decreases within normalized coverage, indicating the presence undercovered regions that can arise as a result of insertions or deletions.

After successful genotyping with KiloSeq, we determined the subset of clones that (i) contained a minimum of one missense mutation; (ii) did not contain any insertions or deletions; (iii) did not contain mutations outside of the ORF; (iii) had unique barcodes; (iv) had sufficient read coverage during KiloSeq to allow for confident genotyping. We re-arrayed this filtered subset of clones (Biomatrix BM5-BC robot, S&P Robotics) into a condensed final library of 40 plates containing 6,548 clones.

### High-throughput yeast-based complementation screen

The yeast-based functional assays were established and validated in our previous study (Sun *et al*, 2016). The mutant alleles of the yeast

temperature-sensitive strains used in this study are *ubc9-2, smt3-331, thi80-ph, and cmd1-1*. The high-throughput screen was performed as follows: the POPCode generated mutant library was transferred to the expression vector pHYCDest (Sun *et al*, 2016) by en masse Gateway LR reactions followed by transformation into NEB5α competent *E. coli* cells (New England Biolabs) and selection for ampicillin resistance.

For the DMS-BarSeq approach, plasmids extracted from a pool of 6,548 barcoded and KiloSeq-validated mutant clones, together with barcoded null and wild-type controls, were transformed into a *S. cerevisiae* strain carrying a temperature-sensitive (ts) allele which can be functionally complemented by the corresponding wild-type human gene (Sun *et al*, 2016). Complexity for this transformation was ~100,000 CFU. For the time series BarSeq screen, the pools were grown separately at both non-selective (25°C) and selective (38°C) temperatures in triplicates to be examined at five different timepoints (0, 6, 12, 24, and 48 h) yielding 30 samples. For each sample, plasmids were extracted from 10 ODU of cells and used as templates for the downstream barcode PCR amplification. The barcode loci were amplified for each library of plasmids with primers carrying sample-specific tags and then sequenced on an Illumina NextSeq 500.

For the DMS-TileSeq approach, plasmids extracted from a pool of ~100,000 clones were transformed into the corresponding *S. cerevisiae* temperature-sensitive strain yielding around 1,000,000 total transformants. Plasmids were prepared from two of 10 ODU of cells and used as templates for the downstream tiling PCR (two replicates of non-selective condition). Two of 40 ODU of cells were inoculated into 200 ml medium and grown to full density with shaking at 36°C and plasmids extracted from 10 ODU of each culture were used as templates for the downstream tiling PCR (two replicates of selective condition). In parallel, plasmid expressing the wild-type ORF was transformed to the corresponding *S. cerevisiae* ts strain and grown to full density under the selection. Plasmids were extracted from two of 10 ODU of cells and used as templates for the downstream tiling PCR (two replicates of wild-type control). For each plasmid library, the tiling PCR was performed in two steps: (i) The targeted region of the ORF was amplified with primers carrying a binding site for Illumina sequencing adaptors; (ii) each first-step amplicon was indexed with an Illumina sequencing adaptor in the second-step PCR. We perform paired-end sequencing on the tiled regions across the ORF.

**Fitness scoring and refinement**

For DMS-BarSeq, a computational pipeline was implemented to identify and count individual sample tags and barcode combinations within each read (see "Code and data availability" section). We then calculate how much better (or worse) each clone grows compared to the pool average, cumulatively across timepoints. To this end, we first calculated the relative population size by dividing each clone's barcode count by the total number of barcodes in each condition. We then calculated the estimated absolute population size for each clone at each timepoint by multiplying the relative population size with the estimated total number of cells on the respective plate at the corresponding timepoint (obtained from OD measurements). We then treat the hourly growth rate between each individual timepoint compared to the pool average as an individual estimate of fitness,

all of which act cumulatively. Formally, this corresponds to the following:

Let $c_{i,t_k}^{(\tau)}$ be the barcode count for clone $i$, timepoint $t_k$ at temperature $\tau$, then $\forall i \in \{1 \leq i \leq N | i \in \mathbb{N}\}$, $\forall k \in \{1 \leq k \leq 5 | k \in \mathbb{N}\}$, $\forall \tau \in \{25°, 37°\}$

$$r_{i,t_k}^{(\tau)} = \frac{c_{i,t_k}^{(\tau)}}{\sum_j c_{j,t_k}^{(\tau)}}$$

$$P_{i,t_k}^{(\tau)} = r_{i,t_k}^{(\tau)} \cdot P_{*,t_k}^{(\tau)}$$

$$\rho_{i,t_k}^{(\tau)} = \sqrt[(t_k - t_{k-1})]{\frac{P_{i,t_k}^{(\tau)}}{P_{i,t_{k-1}}^{(\tau)}}}$$

$$\phi_{i,t_k}^{(\tau)} = \frac{\rho_{i,t_k}^{(\tau)}}{\rho_{*,t_k}^{(\tau)}}$$

$$\phi'_{i,t_k} = \frac{\phi_{i,t_k}^{(37)°}}{\phi_{*,t_k}^{(25)°}}$$

$$s_i = \prod_k \phi'_{i,t_k}$$

$$s'_i = \frac{s_i - s_{\text{null}}}{s_{\text{wt}} - s_{\text{null}}},$$

where $r_{i,t_k}^{(\tau)}$ is the relative population size for clone $i$, timepoint $t_k$ at temperature $\tau$, $P_{i,t_k}^{(\tau)}$ is the absolute population size for clone $i$, timepoint $t_k$ at temperature $\tau$, $\rho_{i,t_k}^{(\tau)}$ is the measured hourly growth rate for clone $i$, timepoint $t_k$ at temperature $\tau$, $\phi_{i,t_k}^{(\tau)}$ is the fitness advantage relative to the pool growth for clone $i$, timepoint $t_k$ at temperature $\tau$, $\phi'_{i,t_k}$ is the normalized relative fitness advantage for clone $i$, timepoint $t_k$, and $s_i$ is the cumulative normalized relative fitness advantage for clone $i$. Finally, $s'_i$ is the fitness score relative to the internal null and wild-type controls, and this results in null-like mutants receiving a score of zero and wild-type-like mutants receiving a score of one.

Given limited amounts of replicates, the empirical standard deviations calculated for each clone or variant can be expected to be imprecise. Baldi and Long (2001) have previously described a method for Bayesian regularization or refinement of the standard deviations which yield more robust estimates, leading to less classification error in statistical tests. Briefly, a prior estimate of the standard deviation is computed by linear regression based on the number of barcodes in the permissive condition and the fitness score. The prior is then combined with the empirical value using Baldi and Long's original formula

$$\sigma^2 = \frac{v_n \sigma_n^2}{v_n - 2} = \frac{v_0 \sigma_0^2 + (n-1)s^2}{v_0 + n - 2}$$

where $v_0$ represents the degrees of freedom assigned to the prior estimate, $\sigma_0$ is the prior estimate resulting according to the regression, $n$ represents the degrees of freedom for the empirical data (i.e., the number of replicates), and $s$ is the empirical standard deviation. The methods were implemented as part of a larger DMS analysis package (see "Code and data availability" section).

For DMS-TileSeq, raw sequencing reads were aligned to the reference ORF cDNA sequences using Bowtie-2 (Langmead & Salzberg, 2012) and a custom Perl script was used to parse and compare the forward and reverse read alignment files to count the number of co-occurrences of a codon change in both paired reads. Mutational counts in each condition were normalized to sequencing depth at the respective position. Variants for which the number of reads in the non-permissive condition was within three standard deviations of the read count in the wild-type control were considered poorly measured and removed. Then, the normalized mutational counts from the wild-type control libraries (control for sequencing errors) were subtracted from the normalized mutational counts from the non-selective and selective conditions, respectively. Finally, the enrichment ratio was calculated for each variant based on the adjusted mutational counts before and after selection.

### Re-scaling of fitness metrics

The results from the barcoded and regional sequencing screens do not scale linearly to each other. We used regression to find a monotonic transformation function $f(x) = a \cdot e^x + b \cdot x + c$ between the two screens' respective scales. The standard deviation is transformed accordingly using a Taylor series-based approximation: $\sigma' = \sigma \cdot (a \cdot e^\mu + b)$. After both datasets have been brought to the same scale, we can join corresponding data points using weighted means, where the weight is inversely proportional to the Bayesian regularized standard error. Output standard error was adjusted to account for differences in input fitness values and increased sample size:

$$w_0 = \frac{1}{1 + \frac{\sigma_{\bar{x}}^{(0)}}{\sigma_{\bar{x}}^{(1)}}} \; ; \; w_1 = \frac{1}{1 + \frac{\sigma_{\bar{x}}^{(1)}}{\sigma_{\bar{x}}^{(0)}}}$$

$$\mu_{\text{joint}} = w_0 \cdot \mu_0 + w_1 \cdot \mu_1$$

$$\sigma_{\text{joint}}^2 = w_0 \cdot \left(\sigma_0^2 + \mu_0^2\right) + w_1 \cdot \left(\sigma_1^2 + \mu_1^2\right) - \mu_{\text{joint}}^2$$

$$\sigma_{\bar{x}}^{(\text{joint})} = \frac{\sigma_{\text{joint}}}{\sqrt{df_0 + df_1}}$$

where $\mu_0$ is the DMS-BarSeq value, $\sigma_0$ the associated standard deviation, $\sigma_{\bar{x}}^{(0)}$ the associated standard error, $df_0$ the associated degrees of freedom, $\mu_1$ is the DMS-TileSeq value, $\sigma_1$ the associated standard deviation, $\sigma_{\bar{x}}^{(1)}$ the associated standard error, and $df_1$ the associated degrees of freedom. These steps were implemented as part of a larger DMS analysis package (see "Code and data availability" section).

### Imputation of missing data

Next, we aimed to find a machine-learning method that would allow us to input the missing parts of the map. The first step toward this was to gather suitable features. We first evaluated the most promising features using linear regression and then applied a random forest model using all the available features.

The most important features were intrinsic, that is, directly derived from unused information in the screen. These are the average fitness across variants at the same position; the average fitness

of multi-mutant clones that contain the variant of interest; the estimated fitness according to a multiplicative model to infer mutant fitness A using a double mutant AB and single mutant B. Another set of features was computed from differences between various chemical properties of the wild-type and mutant amino acids. These properties include size, volume, polarity, charge, and hydropathy. A third set of features is derived from the structural context of each amino acid position. This includes secondary structure, solvent accessibility, burial in interfaces with different interaction partners and involvement in hydrogen bonds or salt bridges with interaction partners. Secondary structures were calculated using Stride (Frishman & Argos, 1995). Solvent accessibility and interface burial were calculated using the GETAREA tool (Fraczkiewicz & Braun, 1998) on the following PDB entries: for UBE2I: 3UIP (Gareau *et al*, 2012); 4W5V (Boucher *et al* unpublished); 3KYD (Olsen *et al*, 2010); 2UYZ (Knipscheer *et al*, 2007); 4Y1L (Alontaga *et al*, 2015); for SUMO1: 2G4D (Xu *et al*, 2006); 2IO2 (Reverter & Lima, 2006); 3KYD (Olsen *et al*, 2010); 3UIP (Gareau *et al*, 2012); 2ASQ (Song *et al*, 2005); 4WJO (Cappadocia *et al*, 2015); 4WJQ (Cappadocia *et al*, 2015); 1WYW (Baba *et al*, 2005); for calmodulin: 3G43 (Fallon *et al*, 2009); 4DJC (Sarhan *et al*, 2012); and for TPK1: 3S4Y (Baker *et al*, 2001).

Hydrogen bond and salt bridge candidates were predicted using OpenPyMol and evaluated for validity by manual inspection. Additional features used are the BLOSUM score for a given amino acid change, the PROVEAN score, and the evolutionary conservation of the amino acid position. Conservation was obtained by generating a multiple alignment of direct functional orthologs across many eukaryotic species using CLUSTAL (Sievers & Higgins, 2014), which was used as input for AMAS (Livingstone & Barton, 1993). We then applied the complete set of features in a random forest model using the R package Random Forest (Breiman, 2001) version 4.6–12 with the default settings for all hyperparameters ($n_{\text{tree}} = 500$, $m_{\text{try}} = n_{\text{feat}}/3$, replace = TRUE, sampsize = $n_{\text{obs}}$, nodesize = 5, maxnodes = NULL, nPerm = 1). These procedures were implemented as part of a larger DMS analysis package (see "Code and data availability" section).

### Refinement of low-confidence measurements

The machine learning predictions resulting generated above can also be used to refine experimental measurements of lower confidence. To this end, the corrected standard error associated with each data point can be used to determine the weight of assigned to the measurement.

$$w_0 = \frac{1}{1 + \frac{\sigma_{\bar{x}}^{(0)}}{\sigma_{\bar{x}}^{(1)}}} \; ; \; w_1 = \frac{1}{1 + \frac{\sigma_{\bar{x}}^{(1)}}{\sigma_{\bar{x}}^{(0)}}}$$

$$\mu_{\text{joint}} = w_0 \cdot \mu_0 + w_1 \cdot \mu_1$$

$$\sigma_{\text{joint}}^2 = w_0 \cdot \left(\sigma_0^2 + \mu_0^2\right) + w_1 \cdot \left(\sigma_1^2 + \mu_1^2\right) - \mu_{\text{joint}}^2$$

$$\sigma_{\bar{x}}^{(\text{joint})} = \frac{\sigma_{\text{joint}}}{\sqrt{df_0 + df_1}}$$

where $\mu_0$ is the measured value, $\sigma_0$ the associated standard deviation, $\sigma_{\bar{x}}^{(0)}$ the associated standard error, $df_0$ the associated degrees of

freedom, $\mu_1$ is the Random Forest-predicted value, $\sigma_1$ the associated standard deviation as approximated by cross-validation RMSD, $\sigma_{\bar{x}}^{(1)}$ the associated standard error, and $df_1$ the associated virtual degrees of freedom. The methods were implemented as part of a larger DMS analysis package (see "Code and data availability" section).

### Experimental validation by yeast spotting assays

To validate the reliability of the fitness scores obtained during the screen, we selected three subsets of clones from our original UBE2I variant library: (i) a set of clones carrying variants with functional scores representing the full spectrum in the screen; (ii) a set of clones carrying hypercomplementing variants in the screen; and (iii) a set of clones carrying variants not present in the imputation training dataset. After genotype verification using Sanger sequencing, each variant was transferred to the yeast expression plasmid pHYCDest by Gateway technology and individually transformed into the yeast ts mutant strain. Cells were grown to saturation in 96-well cell culture plates at room temperature. Each culture was then adjusted to an OD600 of 1.0 and serially diluted to $5^{-1}$, $5^{-2}$, $5^{-3}$, $5^{-4}$, $5^{-5}$, and $5^{-6}$. These cultures (5 µl of each) were then spotted on SC-LEU plates as appropriate to maintain the plasmid and incubated at either the permissive or non-permissive temperatures for 2 days. Each variant was assayed alongside negative and positive controls for loss of complementation (expression of either the wild-type human protein or a GFP control). Results were interpreted by comparing the growth difference between the yeast strains expressing human genes and the corresponding control strain expressing the GFP gene.

Quantification of spotting assay images (for Appendix Fig S3) was performed as follows: Using blinded manual inspection, the following scores were assigned: 0—no colonies visible; 0.25—colonies visible up to the first dilution; 0.25—colonies visible up to the second dilution; 0.75—colonies visible up to the third dilution; 1—colonies visible up to the fourth dilution (a value of 1 was chosen as this corresponds to growth in the wild-type control); 1.25—colonies visible up to the fifth dilution; 1.5—colonies visible up to the sixth dilution.

### Assessing relationship of hyperactive complementation to reversion

To examine whether changing amino acid residues into those residues naturally occur in yeast were more likely to show hyperactive complementation, we compared these cases to changes into residues occurring in other species. The UBE2I amino acid sequence was aligned to that of its orthologs in *S. cerevisiae, D. discoideum,* and *D. melanogaster* using CLUSTAL (Sievers & Higgins, 2014). A custom script was used to extract inter-species amino acid changes and look up the corresponding complementation fitness values in the UBE2I map. Distributions were plotted using the R package beeswarm (Eklund, 2016). The methods were implemented as part of a larger DMS analysis package (see "Code and data availability" section).

### *In vitro* sumoylation comparison

Images from *in vitro* sumoylation assays performed for UBE2I variants by (Bernier-Villamor *et al*, 2002) were scored by visual inspection while blinded to the underlying variant information. Scores were then represented as a heatmap and compared complementation scores from the UBE2I map. The methods were implemented as part of a larger DMS analysis package provided and also available online at https://bitbucket.org/rothlabto/dmspipeline.

### Phylogenetic comparison of different models for hypercomplementation

We used the phydms software package (Bloom, 2017) to test three different models relating the effect of complementation-enhancing substitutions in SUMO1 and UBE2I to actual preference for the substituted amino acid in a real biological context. Specifically, using the substitution models described in (Bloom, 2017), we tested three different ways of relating the evolutionary preference $\pi_{r,a}$ for amino acid a at site r to the fitness score $f_{r,a}$ for this variant. In the first model, $\pi_{r,a} = f_{r,a}$. In the second model, $\pi_{r,a} = \min(f_{r,a},\ f_{r,wt})$ where $f_{r,wt}$ is the fitness score for the wild-type amino acid at site r. In the third model, $\pi_{r,a} = f_{r,a}$, if $f_{r,a} \leq f_{r,wt}$ and $1/f_{r,a}$ otherwise. We fit each of these models to the set of Ensembl homologs with at least 75% sequence identity to the human protein. As shown in Appendix Table S1, in all cases the last model (which assigns low preference to variants that strongly enhance activity) best fits the sequences. The computer code that performs this analysis is available on GitHub at https://github.com/jbloomlab/AtlasPaper_SUMO1_UBE2I_ExpCM.

### Code and data availability

All code associated with this work can be checked out using mercurial from the following repositories: (i) for the KiloSeq analysis pipeline: https://bitbucket.org/rothlabto/kiloseq; (ii) for the POPCode oligo design tool: https://bitbucket.org/rothlabto/popc odesuite; (iii) for the BarSeq sequence analysis pipeline: https://bitbucket.org/rothlabto/screenpipeline; (iv) for the TileSeq sequence analysis pipeline: https://bitbucket.org/rothlabto/tilese q_package; (v) for all raw data and downstream analyses: https://bitbucket.org/rothlabto/dmspipeline. Raw sequencing data can be obtained from the NCBI Short Read Archive, accession numbers SRP109101 (KiloSeq) and SRP109119 (DMS screens). All final variant maps and associated data tables can be found in Dataset EV1 and can be downloaded at http://dalai.mshri.on.ca/~jweile/projects/dmsData/ or from the Biostudies database (www.ebi.ac.uk/biostudies/), accession number S-BSST60. Original data for the *in vitro* sumoylation analysis can be found in Bernier-Villamor *et al* (2002).

**Expanded View** for this article is available online.

### Acknowledgements

The authors thank Amy Caudy, Lincoln Stein, Igor Stagljar, Chris Lima, and Brian Raught for their advice, and thank Brenda Andrews and Charles Boone for kindly providing temperature-sensitive yeast mutant strains. The authors gratefully acknowledge funding by the National Human Genome Research Institute of the National Institutes of Health (NIH/NHGRI) Center of Excellence in Genomic Science (CEGS) Initiative (HG004233), the Canadian Excellence Research Chairs (CERC) Program, and the Ontario Ministry of Research and Innovation (MRI).

## Author contributions

FPR, JW, SS, and AGC conceived the project; SS, AGC, JK, MV, and CW performed the DMS experiments and manual validations; JM, MT, and FR conceived the KiloSeq method, AGC, JK, and JM performed KiloSeq; JW, SS, and NL developed the analysis pipeline; YW and JW developed the machine-learning imputation and refinement method with advice from DF; JW, CP, and PA performed structural and epistasis analyses; SS and FY curated the list of assayable genes; JB performed the evolution analysis; SY and BN helped conduct the blind test with Invitae variant data; GT constructed ts strains; DEH and MV provided human clones; and JW, FPR, and SS wrote the manuscript. FPR supervised the project.

## Conflicts of interest

FPR is a shareholder and scientific advisory board member of SeqWell Inc. and of Ranomics, Inc.. RN and SY are employees of Invitae, Inc.

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
