## [Review Process File · Molecular Systems Biology]

A framework for exhaustively mapping functional missense variants

Jochen Weile, Song Sun, Atina G. Cote, Jennifer Knapp, Marta Verby, Joseph Mellor, Yingzhou Wu, Carles Pons, Cassandra Wong, Natascha van Lieshout, Fan Yang, Murat Tasan, Guihong Tan, Shan Yang, Douglas M. Fowler, Robert Nussbaum, Jesse D. Bloom, Marc Vidal, David E Hill, Patrick Aloy & Frederick P. Roth

Review timeline:

Submission date:	9 August 2017
Editorial Decision:	7 September 2017
Revision received:	18 October 2017
Editorial Decision:	8 November 2017
Revision received:	15 November 2017
Accepted:	18 November 2017

Editor: Maria Polychronidou

Transaction Report:

1st Editorial Decision

7 September 2017

Thank you again for submitting your work to Molecular Systems Biology. We have now heard back from the two referees who agreed to evaluate your study. As you will see below, the reviewers raise a series of concerns, which we would ask you to address in a revision of the manuscript.

The reviewers' recommendations are rather clear so I think that there is no need to repeat the points listed below. Please let me know if you would like to discuss any of these points in further detail.

REVIEWER REPORTS

Reviewer #1:

Comments to the Author:

In the manuscript by Jochen and colleagues, the authors proposed two deep mutational scanning frameworks to test all possible amino acid changes. They applied the method to four proteins in order to identify variants with functional impact. The results identified pathogenic variants and demonstrated the biological relevance of the fitness scores.

Major comments:

1. The author stated that in TileSeq libraries, the multiple amino acid substitutions is not a confounding factor through the distinct score distributions of synonymous variants and nonsense variants in Fig S2. However, the distributions of missense variants are bimodal in UBE21/SUMO1, but unimodal IN TPK1/CALM1, this could still be an indication of confounding factor.
2. The author compared the DMS-BarSeq and DMS-TileSeq results to assess DMS-TileSeq's reliability. It will be clearer if the authors show the same analysis of DMS-TileSeq complementation screen scores compared with structural context as in Supplement S1
3. The author used random forest model to impute missing entries in the map, but the hyper-parameter used in the model is not reported.
4. Different proteins have different missing amino acid change percentage, is the imputation method applicable to all proteins? Can the random forest model be applied to all proteins or need to re-train for each protein?
5. Can the authors give a suggested value of functional scores for pathogenic variants for clinical utility?

Reviewer #2:

Weile et al present an experimental and statistical deep mutational scanning (DMS) pipeline to model variants' protein-level functional impacts. Four human genes were mutagenized and subjected to pooled yeast complementation assays to obtain fitness measurements covering a majority of possible amino-acid substitutions. The authors argue that these estimates serve as measures of variant pathogenicity. A major component of their work is a framework to impute missing values and refine noisy values from the raw DMS data.

This effort is timely in the face of the ever-increasing burden of variant functional interpretation from personal genome sequencing, and it adds to short list of human genes subjected to saturation functional analysis. The presented fitness maps for four genes appear to be substantially complete and high-quality. The measured effects in the yeast screens are well-validated (in yeast), and despite modest accuracy the imputation may be very helpful for often-incomplete DMS maps. The authors relate their maps to the known structures and functions of each gene and make some interesting observations. Despite these strengths, I had concerns about the clarity of the manuscript as well as how it is presented, outlined below.

Major issues

- Variant scoring

o Various types of scores are produced (raw barseq, raw tileseq, joint, imputed, refined) and it is often unclear which one is presented. This should be explicit in the text and figures. Fig S4 shows "regularized" scores. Are those the same as refined scores?

o Does the scoring approach require three replicates at each time point? Depending on the screen, this could be a considerable challenge and should be noted.

o For DMS-barseq, a series of transformations are applied to the raw barcode frequencies. What's the intuition behind these? How do the resulting scores compare with more simple log-ratios or regression-based measures (e.g., enrich2)?

o Related: Figure S2 shows histograms of 'library enrichments' (log phi). What types of scores are these? I assumed they were something along the lines of $\log(f[\text{mut}_i, \text{selected}] / f[\text{mut}_i, \text{unselected}])$ but this should be described.

- Hypercomplementing allele pathogenicity.

o The UBE2I screen reveals a number of alleles which complement better than the human gene (and comparably to the yeast ortholog) - these are nicely validated in Fig S5.

o But, the hypothesis that hypercomplementing alleles in yeast are pathogenic seems underdeveloped. Is there precedent for from other yeast complementation studies of genes? Is overexpression of any of these genes toxic in animal models or mammalian tissue culture? What is the model of pathogenicity (shifted substrate specificity?) - is this gain or loss of function? This

model is hard to evaluate in the context of UBE2I since its protein sequence is ~100% conserved among vertebrates so that there are no fixed variants to compare to these data. There don't seem to be any pathogenic variants in the literature, and these genes are nearly invariant in the population.

- o "We therefore reinterpreted cases of hyperactive complementation in our map as deleterious and repeated the imputation and refinement procedure. This also allowed for more reliable imputed values (reducing cross-validation RMSD from 0.33 to 0.24)." - it was not explained or clear to me the choice to interpret yeast hypercomplementing alleles as deleterious to human affects (or is even involved) the imputation and refinement procedure-please clarify. Doesn't imputation/refinement attempt to model behavior in the yeast screen (where the true fitness for these variants > 1)?

- Imputation

- o How complete do the screen data need to be for imputation to provide some benefit? Show how model performance decays with increasing sparsity and/or noise among the input data.
- o How much better is this model than one trained only on conservation + physicochemical + structural features?
- o As further justification for building fitness maps on all amino-acid substitutions (as the authors argue for later in the paper), how does imputation perform when trained only on amino acid substitutions reachable by 1-bp mutations, versus all aa substitutions?
- o Although it seems clear that this model has some predictive value, in practice its average error (~0.3 across the four proteins) appears to be much larger than the difference between variant sets the authors' model seeks to discriminate (e.g., in Figure 5A, B). Although this approach may be quite useful for noisy/incomplete DMS maps, its relative noisiness (at least for the difficult task of identifying pathogenic alleles) should be noted as a remaining challenge.
- o The refinement procedure is intuitively appealing, but it does not appear to change the fitness estimates very much (Table 1 - "Refinement > 0.05" column). Further, it was unclear whether the changes actually improve fitness estimates. Table 1 contains a pair of columns "Experimental max(stderr)" and "Refined max(stderr)" which might show this, but they're not defined, and if I am inferring their meaning right, they seem to be based on the max per-variant error when a more reasonable evaluation would be based upon overall error (like RMSE) or classification error. The benefits of this refinement model would be more convincingly demonstrated by showing, eg, (1) improved concordance with validation data, (2) reduced cross-validation error, or (3) improved concordance with actual screen data when applied to those data with synthetic noise spiked in.

- Presentation issues -

- o The suitability of this approach to classify human disease variants holds some promise but overstated in its current form. In particular, the error between replicates in the screen (and during imputation) both are much greater than the apparent difference in scores between bona fide pathogenic and nonpathogenic alleles. Part of this is that among the selected genes, there are very known examples of likely pathogenic and likely neutral variants (and only for TPK1 and calmodulin).
- o Classification performance on the variants in in Table 2 (Invitae cardiac vs non-cardiac indication) appears to have been very sensitive to the choice of threshold, and benefited from a new category ("uncertain") with two variants. This is an unfairly optimistic way to evaluate the DMS performance - appropriate instead would be to score by prAUC as done in the gnomAD vs ClinVar comparison. Table 2 should also show values from bioinformatics predictions and some discussion of whether DMS improved classification in this variant set as it did for ClinVar vs gnomAD variants. (And, what do 'mild' and 'prereg' in Table 2 mean?)
- o UBE2I and SUMO1 should be plotted separately rather than aggregated in Figure 5A. Similar plots should be shown for TPK1 and CALM1/2/3. And, what is the premise for this comparison - are UBE2I and SUMO1 essential and dosage-sensitive? Are cancer somatic variants expected to be depleted for LOF?
- o While powerful, a few other limitations on this approach should be noted: it appears to be restricted to modeling loss of function mutations, and only those with protein-level effects (important as many pathogenic coding variants impact splicing). Finally, it requires that a yeast allele exist and human gene complement it (<= 5% of human genes), with strength of complementation in yeast negatively correlated with pathogenicity of human variant. The authors are commended for their census of potentially DMS-able human genes in Table S3 (they estimate 57%) but it should be noted that these DMS experiments in human cells are considerably lower-throughput and more expensive with current approaches.

- Performance of mutagenesis and readout methods

- o The authors develop two mutagenesis readout methods - barseq and tileseq. The advantages, drawbacks, costs, and required equipment for each should be more clearly stated (perhaps in a table), to better inform a reader considering adopting one or the other. On a related note, the requirement for picking and arrayed handling of barseq clone libraries should be made more explicit in the method's description, since this a major barrier to adoption by many labs.
- o If a lab has access to the robotics required to perform barseq, is tileseq still a better choice (as suggested by the choice to go with tileseq for the remaining genes)?
- o A major drawback of tileseq will be its performance on sequences longer than the (mean) ~160 aa cDNAs selected, where (1) individual mutations will be less frequent and closer to the background rate of sequencing errors, (2) a larger majority of each tile will be wildtype, and (3) libraries become dominated by multi-mutant clones, which tileseq cannot resolve. It is asserted in the discussion (pages 9-10) that tileseq will address DMS of longer genes than the "less than 200 amino acids of previous DMS - I think those claims are unsupported (and, notably, 3 of 4 genes studied here are <160aa). And, in any event, overlapping paired-end sequencing for DMS variant counting is not novel and shouldn't be sold as such.
- o The authors mention a frequency threshold, "we only conserved those variants present with 'allele frequency' sufficient to allow confident detection..." (what threshold was used?). The authors state that they detect 2563/3012 possible aa substitutions by tileseq, but previously using their clone-resolved barseq, they stated that their library included only 1848/3012. I thought I understood that this was the same library (pooled from arrayed clones) - is this incorrect? If not, how is tileseq picking up these extra clones?
- o The characteristics of the authors' POPcode approach are not shown. How many clones are have 0, 1, 2 programmed mutations? How many have additional non-programmed mutations? What is the distribution of mutagenesis over the protein sequence? How uniformly are these mutations represented at the achieved depth of transformant pool size or in smaller samplings of this library?
- o Authors claim that most base-calling errors can be eliminated by sequencing with overlapping paired ends. This should be shown, e.g., by displaying the mutation rate from these reads in non-mutant library, across the amplicon and in aggregate.
- o The validation shown in Fig 1B (lower panel) of barseq data, showing consistency between independent clones with the same mutation, is compelling. (Does this include single mutant clones only or are averaged out multi-mutant effects here?). I realize this is more difficult to show for tileseq, but one option would be to show consistency between different codon mutations leading to the same amino acid change under tileseq?
- o Minor - page 4, "In the TileSeq libraries, some clones will necessarily contain multiple amino acid substitutions" is a bit misleading -- it is the majority rather than 'some'.
- o TileSeq can't explicitly model the wildtype sequence since it does not carry a specific marker. When barseq data aren't available, how are scores calibrated to wildtype? Via synonymous variants?
- o POPcode mutagenesis strategy is probably not suitable for longer genes where libraries become dominated by multi-mutation clones. This is less of a problem for the short genes selected here, but e.g., at the given mutagenesis rate, for a 500-aa gene, <1% of mutant clones will have a single amino acid mutation. Given the availability other methods which create single-mutant libraries this limitation should be noted.
- o Randomized mutagenesis using oxidized nucleotides is mentioned in the methods section but it wasn't clear if/where this technique was used. Was this how the other three mutant libraries were constructed? If so, how did they reach mutational coverage comparable to that of POPCode?

Minor issues

- Figure 1B top vs bottom is unclear. Bottom appears to compare estimates from different barcodes carrying the same variant within the same selection experiment. What does top panel show? Separate selections starting from the same transformant pool (each with per-timepoint triplicate as described in methods)? Or are these simply replicated sequencing of the same selection? If the latter, it would seem most of the variability comes from the selection itself, not the readout.
- Fig S2. Is wildtype=0 here?
- Fig S7. Please provide a legend for individual variants (I assumed green = population, i.e., GnomAD or 1000GP, red=pathogenic ClinVar, blue=???). Is the upper panel from GnomAD (haploid alleles) and the bottom panels from 1000 Genomes? To be clear, those are different variant sets which should be noted.
- Quantification of validation in Fig S4 is unclear. How are spotting assays quantified? Spearman

correlation versus what appear to be discrete spotting assay scores with 6 values might be misleading. What about R^2 , or auPR, thresholding on the spotting-based scores? Are imputed score error bars simply the 10xCV RMSD for the entire library or do they have a point-specific meaning? What does "high-quality" (in the legend) mean?

- The description of KIL0seq (page 21 of supplementary methods) is vague, and obscures some of the method's limitations. Crucially it appears to require as input arrayed clone libraries. It also requires well-by-well construction of shotgun libraries from these clones, which is somewhat facilitated by the SeqWell approach, but still requires a single well for each clone. As far as I could tell, this produces reads with a well-specific tag (to identify clone of origin) and a shotgun read derived randomly from the substrate (vector backbone, cloned mutant library, or mutant barcode). These would then be split by well tag and assembled. I couldn't figure out how this would generate pairs of reads where "one read will reveal the well tag and the barcode locus, whereas the other will contain a fragment of the mutant ORF". A schematic figure might help here.

- The epistasis analysis in Fig S12 is very interesting but I don't think it was mentioned in the text anywhere. It was surprising to see that most pairwise mutations showing significant epistasis were not near each other. Were any other examples of genetically interacting mutations validated (beyond the I4/P69S example shown) to bolster this finding?

- Page 9, "We see three arguments" - four arguments are listed.

1st Revision - authors' response

18 October 2017

Reviewer #1:

In the manuscript by Jochen and colleagues, the authors proposed two deep mutational scanning frameworks to test all possible amino acid changes. They applied the method to four proteins in order to identify variants with functional impact. The results identified pathogenic variants and demonstrated the biological relevance of the fitness scores.

Major comments:

1. The author stated that in TileSeq libraries, the multiple amino acid substitutions is not a confounding factor through the distinct score distributions of synonymous variants and nonsense variants in Fig S2. However, the distributions of missense variants are bimodal in UBE21/SUMO1, but unimodal IN TPK1/CALMI, this could still be an indication of confounding factor.

We thank the reviewer for this comment. As the reviewer notes, our argument that measurements of missense variant function at a given position are not confounded by the presence of variants at other positions in the clone was based on the observation that the score distributions for synonymous and nonsense variants were quite distinct (whereas confounding would tend to move the synonymous distribution towards the nonsense distribution). Having established that our results are not consistent with a major confounding effect from variants at other positions, we interpret the shape of the score distribution of all missense variants for each gene as suggesting that there is a continuum of functional impact for missense variants. The bimodality of missense variants for *UBE21* and *SUMO1* suggests that most variants lie at one end or the other of this continuum, while the shape for *TPK1* suggests that functional impact tends more towards deleteriousness, and for *CALMI* tends more towards neutrality. We have updated the results and discussion sections to reflect this interpretation.

2. The author compared the DMS-BarSeq and DMS-TileSeq results to assess DMS-TileSeq's reliability. It will be clearer if the authors show the same analysis of DMS-TileSeq complementation screen scores compared with structural context as in Supplement S1.

Great suggestion. Supplementary Figure S1 (now Appendix Figure S1) has been expanded to also evaluate DMS-TileSeq.

3. The author used random forest model to impute missing entries in the map, but the hyperparameter used in the model is not reported.

We now state that we used default settings for all hyperparameters, and these settings are now explicitly stated in the methods section.

4. Different proteins have different missing amino acid change percentage, is the imputation method applicable to all proteins? Can the random forest model be applied to all proteins or need to re-train for each protein?

The RandomForest predictor is re-trained separately for each map. We have edited the main text (page 7) to make this more clear. Training the imputation predictor on multiple maps may improve performance in the future, and we appreciate the suggestion. We have not yet explored this idea, given that there may be challenges getting perfectly comparable linear scaling of scores across maps, but we now raise this as a future direction in the Discussion.

5. Can the authors give a suggested value of functional scores for pathogenic variants for clinical utility?

Pages 8 and 9 of the manuscript discuss the potential applicability of our approach to re-classifying to variants of unknown significance, which would be of clinical utility wherever genetic diagnosis is actionable, or where genetic diagnosis leads to stratification of patients according to genetic cause. Although the latter will not generally have immediate clinical utility, it could identify cryptic disease types that are ultimately found to benefit from distinct therapies.

Reviewer #2:

Weile et al present an experimental and statistical deep mutational scanning (DMS) pipeline to model variants' protein-level functional impacts. Four human genes were mutagenized and subjected to pooled yeast complementation assays to obtain fitness measurements covering a majority of possible amino-acid substitutions. The authors argue that these estimates serve as measures of variant pathogenicity. A major component of their work is a framework to impute missing values and refine noisy values from the raw DMS data.

This effort is timely in the face of the ever-increasing burden of variant functional interpretation from personal genome sequencing, and it adds to short list of human genes subjected to saturation functional analysis. The presented fitness maps for four genes appear to be substantially complete and high-quality. The measured effects in the yeast screens are well-validated (in yeast), and despite modest accuracy the imputation may be very helpful for often-incomplete DMS maps. The authors relate their maps to the known structures and functions of each gene and make some interesting observations. Despite these strengths, I had concerns about the clarity of the manuscript as well as how it is presented, outlined below.

We appreciate the thorough and constructive review, and hope the reviewer will agree that we have addressed the issues raised.

Major issues

1. Variant scoring

1.a. Various types of scores are produced (raw barseq, raw tileseq, joint, imputed, refined) and it is often unclear which one is presented. This should be explicit in the text and figures. Fig S4 shows "regularized" scores. Are those the same as refined scores?

Thank you. With hindsight we agree that we could have better distinguished the types of scores ("regularized" was our previous in-house term for "refined", and we failed to correct this occurrence of it). We have revised the text and figures to more clearly indicate the type of score used in each context.

1.b. Does the scoring approach require three replicates at each time point? Depending on the screen, this could be a considerable challenge and should be noted.

Replicate plates are processed in parallel and thus the added cost for a third replicate was negligible relative to the total cost of the experiment. The majority of the time taken by the screening protocol is spent waiting for cells to grow, so that additional replicates do not substantially increase the screening time. It may well be true that we have performed more replicates than were needed, but we have not addressed that here.

1.c. For DMS-barseq, a series of transformations are applied to the raw barcode frequencies.

What's the intuition behind these? How do the resulting scores compare with more simple log-ratios or regression-based measures (e.g., enrich2)?

We expanded on the description of the scoring scheme in the Methods section (page 17) to better convey the intuition behind each step. In short, we calculate how much better (or worse) each clone grows compared to the pool average, cumulatively across timepoints.

Before settling on our cumulative fitness-advantage score, we tried other scoring methods including log ratios, area under growth curve, regression using logistic growth models. We found that cumulative fitness-advantage score gave slightly better agreement between biological replicates in terms of Pearson correlation coefficient, but we make no claims of superiority of the scoring scheme. Although we did examine regression using logistic growth models, and we now cite it as an exemplar of such methods, we did not specifically examine Enrich2, which had not been published yet at the time we needed to settle on a scoring scheme.

1.d Related: Figure S2 shows histograms of 'library enrichments' (log phi). What types of scores are these? I assumed they were something along the lines of $\log(f[\text{mut}_i, \text{selected}] / f[\text{mut}_i, \text{unselected}])$ but this should be described.

Thank you. We have improved the axis labels on Figure S2 (now Appendix Figure S7) and expanded on its caption to make the meaning of the score more clear.

2. Hypercomplementing allele pathogenicity.

2.a. The UBE2I screen reveals a number of alleles which complement better than the human gene (and comparably to the yeast ortholog) - these are nicely validated in Fig S5. But, the hypothesis that hypercomplementing alleles in yeast are pathogenic seems underdeveloped. Is there precedent for from other yeast complementation studies of genes? Is overexpression of any of these genes toxic in animal models or mammalian tissue culture? What is the model of pathogenicity (shifted substrate specificity?) - is this gain or loss of function? This model is hard to evaluate in the context of UBE2I since its protein sequence is ~100% conserved among vertebrates so that there are no fixed variants to compare to these data. There don't seem to be any pathogenic variants in the literature, and these genes are nearly invariant in the population.

We agree that this would be a great direction for expanded inquiry. Given the great challenges to such an inquiry (e.g., a shortage of known pathogenic variants for the maps we provide here, as the reviewer notes), we respectfully suggest that this would go beyond the scope of the current study. We are excited to explore this direction further, as additional DMS maps emerge.

2.b. "We therefore reinterpreted cases of hyperactive complementation in our map as deleterious and repeated the imputation and refinement procedure. This also allowed for more reliable imputed values (reducing cross-validation RMSD from 0.33 to 0.24)." - it was not explained or clear to me the choice to interpret yeast hypercomplementing alleles as deleterious to human affects (or is even involved) the imputation and refinement procedure-please clarify. Doesn't imputation/refinement attempt to model behavior in the yeast screen (where the true fitness for these variants > 1)?

The decision to treat hypercomplementing variants in the tested genes as deleterious is primarily based on the phylogenetic analysis described on page 6, which suggest that there is an evolutionary disadvantage in humans (or at least species closely related to humans) associated with these variants. We re-trained the Random Forest model on the revised matrix that inverted the scores of hypercomplementing variants. Cross-validation of the retrained models on the revised matrices showed better (lower) RMSD than the original model on the original scores, suggesting that the revised matrix is more consistent with the corresponding features. We have revised the description in the text to make this more clear.

3. Imputation

3.a. How complete do the screen data need to be for imputation to provide some benefit? Show how model performance decays with increasing sparsity and/or noise among the input data.

Thank you. We have tested the performance with respect to different degrees of map completeness and found the method to be robust. We have added the new Appendix Figure S4 to address this question as well as a new paragraph in the Results section (pages 5-6) to describe the results of the analysis.

3.b. How much better is this model than one trained only on conservation + physicochemical + structural features?

The revised manuscript shows this in Figure 2D. Removal of the information provided by the two main intrinsic features (positional averages and multi-mutant averages) lead to 49% and 40% increase in the mean squared error of the prediction, respectively.

3.c. As further justification for building fitness maps on all amino-acid substitutions (as the authors argue for later in the paper), how does imputation perform when trained only on amino acid substitutions reachable by 1-bp mutations, versus all aa substitutions?

Thank you. The new Appendix Figure S4 to address this question in addition to the analysis described in comment 3a. As a benchmark, Appendix Figure S4 also shows the performance of regression-based imputation using PolyPhen2 scores. We have also added an accompanying description in the Results section (page 6).

3.d. Although it seems clear that this model has some predictive value, in practice its average error (~0.3 across the four proteins) appears to be much larger than the difference between variant sets the authors' model seeks to discriminate (e.g., in Figure 5A, B). Although this approach may be quite useful for noisy/incomplete DMS maps, its relative noisiness (at least for the difficult task of identifying pathogenic alleles) should be noted as a remaining challenge.

It should be noted that 'wet-lab' experimental fitness assays have a similar level of disagreement between biological replicates. Considering this, we respectfully argue that our prediction method is surprisingly accurate. Moreover, as demonstrated for CALM1, the resulting scores are able to separate pathogenic variants from other random variants with ~3-fold higher sensitivity than current computational approaches using the same stringent (90%) precision threshold.

3.e. The refinement procedure is intuitively appealing, but it does not appear to change the fitness estimates very much (Table 1 - "Refinement > 0.05" column). Further, it was unclear whether the changes actually improve fitness estimates. Table 1 contains a pair of columns "Experimental max(stderr)" and "Refined max(stderr)" which might show this, but they're not defined, and if I am inferring their meaning right, they seem to be based on the max per-variant error when a more reasonable evaluation would be based upon overall error (like RMSE) or classification error. The benefits of this refinement model would be more convincingly demonstrated by showing, eg, (1) improved concordance with validation data, (2) reduced cross-validation error, or (3) improved concordance with actual screen data when applied to those data with synthetic noise spiked in.

The caption for Table 1 now better describes the displayed quantities. Also, we have added additional text to the main manuscript (page 6) and a new Appendix Figure S6 to show the improved concordance with validation data. This figure shows that, in each case examined, refinement 'nudged' the score in the right direction, but not very far, suggesting that our refinement may have been overly conservative. Although we are keenly interested in exploring alternative strategies for weighting the refinement that are less conservative, we do not think that we yet have enough of the right kind of data to evaluate these alternatives, so that our preference is to stay with the current conservative refinement strategy for this study.

4. Presentation issues

4.a. The suitability of this approach to classify human disease variants holds some promise but overstated in its current form. In particular, the error between replicates in the screen (and during imputation) both are much greater than the apparent difference in scores between bone fide pathogenic and nonpathogenic alleles. Part of this is that among the selected genes, there are very known examples of likely pathogenic and likely neutral variants (and only for TPK1 and calmodulin).

As the reviewer notes, the number of known structural variants prevented us from judging the value of our DMS maps in identifying pathogenic variants. However, there were sufficiently many known pathogenic variants did allow confident comparisons to be made for calmodulin. We respectfully suggest that any impact of systematic and random errors in our data for calmodulin will have already been captured in our precision-recall analysis of our ability to identify pathogenic variants (Figure 5C).

Whether or not the performance we achieved is sufficient for clinical utility is an open and very important question. Precision in practice depends on the prior probability that the variant examined is pathogenic, which in turn depends, for example, on symptoms and family history of the patient, the number of genes associated with the disease, and the background level of non-pathogenic variants in the gene that harbors the target variant. Just as the American College of Medical Geneticists recommends for computational predictions [Richards *et al.*, *Genetics in Medicine* 2015], we expect that estimates of deleteriousness from DMS maps will be used in combination with other evidence sources. We note that our calmodulin map achieved ~3 times the sensitivity of

computational predictors at the same 90% precision threshold (using the same test set with the same prior probability of pathogenicity).

4.b. Classification performance on the variants in in Table 2 (Invitae cardiac vs non-cardiac indication) appears to have been very sensitive to the choice of threshold, and benefited from a new category ("uncertain") with two variants. This is an unfairly optimistic way to evaluate the DMS performance - appropriate instead would be to score by prAUC as done in the gnomAD vs ClinVar comparison. Table 2 should also show values from bioinformatics predictions and some discussion of whether DMS improved classification in this variant set as it did for ClinVar vs gnomAD variants. (And, what do 'mild' and 'prereg' in Table 2 mean?)

The thresholds used to arrive at the classification results in Table 2 were data-driven; more specifically, they were derived by optimizing precision vs recall performance against the known variants from ClinVar and GnomAD. We consider the choice of thresholds to be fair, given that the evaluation against the Invitae indication was performed as a blind test: Only after we had performed our classification based on the optimized thresholds did Invitae reveal the underlying patient indications. Moreover, our statistical test showing correlation of our DMS results with the clinical indication for the Invitae test was based only on our DMS scores (or more specifically the rank according to these scores) and did not depend in any way on our the chosen thresholds.

We have expanded on our description of the thresholding procedure and our statistical test to make each of these points more clearly.

4.c. UBE2I and SUMO1 should be plotted separately rather than aggregated in Figure 5A. Similar plots should be shown for TPK1 and CALM1/2/3. And, what is the premise for this comparison - are UBE2I and SUMO1 essential and dosage-sensitive? Are cancer somatic variants expected to be depleted for LOF?

UBE2I is likely essential in humans (null mutations in mice are embryonic lethal, and genome-scale CRISPR-Cas9 screens show strong fitness defects in many cell lines [Hart *et al*, 2015, Wang *et al*, 2014]. Null mutations in SUMO1 are also embryonic-lethal in mice. Although SUMO1 does not turn up as a fitness gene in genome-scale CRISPR-Cas9 screens [Hart *et al*, 2015, Wang *et al*, 2014], it does seem to dosage sensitive [Alkuraya *et al*, *Science* 2006]. Therefore, we did expect that LoF variants would be depleted for both genes in cancer somatic variants. We aggregated the variants for UBE2I and SUMO1 because there were too few for each alone to support this analysis. We do not draw any strong conclusions from this analysis, other than that our DMS scores behave as expected in this admittedly indirect way. We would be happy to relegate this less-important analysis to the Supplement/Appendix, or to remove it altogether, at the Editor's discretion.

4.d. While powerful, a few other limitations on this approach should be noted: it appears to be restricted to modeling loss of function mutations, and only those with protein-level effects (important as many pathogenic coding variants impact splicing). Finally, it requires that a yeast ts allele exist and human gene complement it ($\leq 5\%$ of human genes), with strength of complementation in yeast negatively correlated with pathogenicity of human variant. The authors are commended for their census of potentially DMS-able human genes in Table S3 (they estimate 57%) but it should be [noted] that these DMS experiments in human cells are considerably lower-throughput and more expensive with current approaches.

We agree that functional complementation will tend to identify only loss-of-function variants, except where there is a known and readily assayable phenotype for gain-of-function variants.

We also agree that yeast will not be a good system to evaluate variant impacts on splicing.

Although each of our assays used a temperature-sensitive yeast mutant, our previous work (Sun *et al.*, *Genome Res.*, 2016) showed that the same complementation assay yields the same result in a null background, so that a temperature-sensitive allele is not required. DMS in a null background could be carried out, for example, using expression of a 'covering wild-type allele' of the relevant yeast gene under an inducible promoter that could be turned off to commence the selection period.

We certainly agree that there are challenges associated with carrying out DMS experiments in human cells, but note that there have been a combination of technological advances that together suggest that this will be coming sooner rather than later. In addition to the availability of CRISPR-

Cas9 to generate homozygous disruptions in target genes, recent advances in ‘landing pad’ technology (Matreyek *et al.*, *NAR*, 2017) now allow integration of a specific sequence into 1-8% of a population of transfected Hek293T cells. Thus, transfection of as few as 1M cells could in theory allow expression of enough integrated variant clones to carry out DMS.

The revised manuscript now covers each of these points in the Discussion section (page 10-12).

5. Performance of mutagenesis and readout methods

5.a. The authors develop two mutagenesis readout methods - barseq and tileseq. The advantages, drawbacks, costs, and required equipment for each should be more clearly stated (perhaps in a table), to better inform a reader considering adopting one or the other. On a related note, the requirement for picking and arrayed handling of barseq clone libraries should be made more explicit in the method's description, since this a major barrier to adoption by many labs.
We have added a new Appendix Table (S2) to compare DMS-BarSeq and DMS-TileSeq, including DMS-BarSeq’s requirement for robotic infrastructure.

5.b. If a lab has access to the robotics required to perform barseq, is tileseq still a better choice (as suggested by the choice to go with tileseq for the remaining genes)?

We ourselves are likely to only use TileSeq going forward (even though we have the robotic infrastructure to use BarSeq), but we now explicitly discuss that the choice between DMS-BarSeq and DMS-TileSeq is very much dependent on the goal of the study. While DMS-TileSeq is preferable for producing a DMS map for single missense variants, DMS-BarSeq is preferable where users wish to examine the behaviour of double-mutants or higher level multi-mutants, e.g. to study intragenic epistasis. It is also more suitable where it is expected that follow-up experiments will be performed on many specific individual variants, as the arrayed library makes it easy to retrieve the corresponding clones. Appendix Table S2 now highlights these differences.

5.c. A major drawback of tileseq will be its performance on sequences longer than the (mean) ~160 aa cDNAs selected, where (1) individual mutations will be less frequent and closer to the background rate of sequencing errors, (2) a larger majority of each tile will be wildtype, and (3) libraries become dominated by multi-mutant clones, which tileseq cannot resolve. It is asserted in the discussion (pages 9-10) that tileseq will address DMS of longer genes than the "less than 200 amino acids of previous DMS - I think those claims are unsupported (and, notably, 3 of 4 genes studied here are <160aa). And, in any event, overlapping paired-end sequencing for DMS variant counting is not novel and shouldn't be sold as such.

We have indeed focused on shorter genes in this study. However, thanks to its ability to fine-tune the underlying mutation rate, POPCode should scale very well to longer genes as well. The oligo-targeted nature of POPCode also allows mutagenesis to be performed separately for each tile, which could serve to allow a high desired allele frequency at mutagenized positions while still permitting a low overall number of mutants per clone. The Discussion section on page 10 now describes this solution.

5.d. The authors mention a frequency threshold, "we only conserved those variants present with 'allele frequency' sufficient to allow confident detection..." (what threshold was used?).

We removed variants for which the number of reads was within three standard deviations of the read count of the wildtype control. We have added this information to the description in the Methods section (page 18).

The authors state that they detect 2563/3012 possible aa substitutions by tileseq, but previously using their clone-resolved barseq, they stated that their library included only 1848/3012. I thought I understood that this was the same library (pooled from arrayed clones) - is this incorrect? If not, how is tileseq picking up these extra clones?

Although POPCode was used to generate both DMS-BarSeq and DMS-TileSeq libraries for UBE2I, this was done separately (and the DMS-BarSeq library was supplemented with some clones from oxidized nucleotide mutagenesis). While the DMS-BarSeq library underwent colony picking and arraying, which limited the total number of clones and thus mutations, the DMS-TileSeq library was limited primarily by the number of independent yeast transformants (~1,000,000 transformants for ~100,000 variant clones). This is described in the revised Methods section (page 15-16).

5.e. The characteristics of the authors' POPcode approach are not shown. How many clones are have 0, 1, 2 programmed mutations? How many have additional non-programmed mutations? What is the distribution of mutagenesis over the protein sequence? How uniformly are these mutations represented at the achieved depth of transformant pool size or in smaller samplings of this library?
Thank you for this suggestion. We have added a new supplementary text and figure to the appendix (ST1) to discuss the properties of the mutagenesis products in detail.

5.f. Authors claim that most base-calling errors can be eliminated by sequencing with overlapping paired ends. This should be shown, e.g., by displaying the mutation rate from these reads in non-mutant library, across the amplicon and in aggregate.

Thank you for this comment, which led us to realize that we had not cited previous works where this approach was used. We now cite Fowler *et al.*, *Nat.Meth.*, 2010 and Whitehead *et al.*, *Nat.Biotech.*, 2012 as previous applications, as well as Zhang *et al.*, *BMC Genomics*, 2016, an in-depth evaluation of the method by which demonstrates the reduction in base-calling error.

5.g. The validation shown in Fig 1B (lower panel) of barseq data, showing consistency between independent clones with the same mutation, is compelling. (Does this include single mutant clones only or are averaged out multi-mutant effects here?)

The lower panel of Figure 1B shows fitness scores for biological replicate clones, that is, clones that carry the exact same sequence but have been tagged with different barcodes. No averaging across clones has taken place. The analysis may have included some cases where there were two independent clones with different barcodes and precisely the same combination of 2 or more mutants, but these will have been negligibly rare.

I realize this is more difficult to show for tileseq, but one option would be to show consistency between different codon mutations leading to the same amino acid change under tileseq?

As an alternative validation, we show the consistency of DMS-TileSeq with low-throughput complementation spotting assays in Appendix Figure S3.

5.h. Minor - page 4, "In the TileSeq libraries, some clones will necessarily contain multiple amino acid substitutions" is a bit misleading -- it is the majority rather than 'some'.

Thank you. We have corrected this statement in the text. (To address issues expressed in another comment, the paragraph in question has also been moved to the discussion section instead and is can now be found on page 11.)

5.i. TileSeq can't explicitly model the wildtype sequence since it does not carry a specific marker. When barseq data aren't available, how are scores calibrated to wildtype? Via synonymous variants?

Correct. The median score of synonymous variants is used to calibrate to wildtype, and the median score of nonsense mutations is also used to anchor the scale for null mutations. We have now made this more clear in the text (page 7).

5.j. POPcode mutagenesis strategy is probably not suitable for longer genes where libraries become dominated by mutli-mutation clones. This is less of a problem for the short genes selected here, but e.g., at the given mutagenesis rate, for a 500-aa gene, <1% of mutant clones will have a single amino acid mutation. Given the availability other methods which create single-mutant libraries this limitation should be noted.

As mentioned above, we have indeed focused on shorter genes in this study. However, thanks to its ability to fine-tune the underlying mutation rate, POPCode scales very well to longer genes as well. Moreover, sub-pools of mutagenic oligos could be used to target mutagenesis to a single tile at a time, which could serve to allow a high desired allele frequency at mutagenized positions while still permitting a low overall number of mutants per clone. We now cover this in the Discussion section (page 10-11)

5.k. Randomized mutagenesis using oxidized nucleotides is mentioned in the methods section but it wasn't clear if/where this technique was used. Was this how the other three mutant libraries were constructed? If so, how did they reach mutational coverage comparable to that of POPCode?

Oxidized nucleotide PCR was originally used to supplement the DMS-BarSeq library for UBE2I, but was not used for any of the other screens, after it became clear that it was not necessary and offered no benefit compared to using POPCode alone. Despite this, some clones in the UBE2I

BarSeq library were derived from oxidized nucleotide mutagenesis. It is mentioned in the methods only for the sake of completeness.

Minor issues

1. *Figure 1B top vs bottom is unclear. Bottom appears to compare estimates from different barcodes carrying the same variant within the same selection experiment. What does top panel show? Separate selections starting from the same transformant pool (each with per-timepoint triplicate as described in methods)? Or are these simply replicated sequencing of the same selection? If the latter, it would seem most of the variability comes from the selection itself, not the readout.*

Thank you. We have clarified the figure caption to address this issue. The top panel shows the correlation between results from biological replicate selections at each time point. These underwent selection in parallel and were sequenced on the same Illumina run with distinct multiplexing indices.

2. *Fig S2. Is wildtype=0 here?*

The x-axis is $\log(\phi)$, that is, the log of the ratio of variant frequency in the selective condition to the variant frequency in the permissive condition. Wild-type variants had similar frequencies in both permissive and non-permissive conditions leading to a $\log(\phi)$ value around 0. That may seem like a long-winded way to say “yes”, but we wanted to note that it is possible in other situations (e.g., where there is a greater abundance of deleterious variants for the wild-type frequency) to change after selection (i.e., to have $\log(\phi)$ differing from 0).

3. *Fig S7. Please provide a legend for individual variants (I assumed green = population, i.e., GnomAD or 1000GP, red=pathogenic ClinVar, blue=??). Is the upper panel from GnomAD (haploid alleles) and the bottom panels from 1000 Genomes? To be clear, those are different variant sets which should be noted.*

We apologize for the missing legend. It has been added. We have also expanded on the caption to make the distinction between the panels more clear. In the top panel, green indeed corresponds to GnomAD variants, while red and blue refer to pathogenic and benign variants from ClinVar, respectively. In the bottom panel, green corresponds to virtual diploid scores assigned to individuals from the 1000 genomes project. This is based on phasing information to determine whether each case was homozygous, heterozygous, or compound heterozygous. All cases were found to be heterozygous. To emulate the recessive nature of the disease, the scores reflect the maximum value across the two alleles. Red and blue similarly correspond to the diploid scores for the cases behind the variant reports in ClinVar. Consulting the literature referenced for each variant, revealed all four of the five cases to be homozygous and one to be compound heterozygous.

4. *Quantification of validation in Fig S4 is unclear. How are spotting assays quantified? Spearman correlation versus what appear to be discrete spotting assay scores with 6 values might be misleading. What about R^2 , or auPR, thresholding on the spotting-based scores? Are imputed score error bars simply the 10xCV RMSD for the entire library or do they have a point-specific meaning? What does “high-quality” (in the legend) mean?*

We have expanded on the figure legend and described the quantification procedure in the methods section to address these questions. In short, the quantification was performed by manual inspection of spotting assay images while blinded to the underlying phenotypes, with scores assigned according to the highest dilution step for which colonies were still visible.

Indeed, the error bars for the imputation are simply the 10x cross-validation RMSD. We show Spearman correlation instead of Pearson correlation due to the non-linear relationship between the spotting assay scores and the screen scores.

The “fraction of missense variants with a high-quality score” in the legend referred to those variants that passed the filter criteria described in the methods section. We have updated the legend text to make this more clear.

5. *The description of KILoSeq (page 21 of supplementary methods) is vague, and obscures some of the method's limitations. Crucially it appears to require as input arrayed clone libraries. It also requires well-by-well construction of shotgun libraries from these clones, which is somewhat facilitated by the SeqWell approach, but still requires a single well for each clone. As far as I could tell, this produces reads with a well-specific tag (to identify clone of origin) and a shotgun read derived randomly from the substrate (vector backbone, cloned mutant library, or mutant barcode).*

These would then be split by well tag and assembled. I couldn't figure out how this would generated pairs of reads where "one read will reveal the well tag and the barcode locus, whereas the other will contain a fragment of the mutant ORF". A schematic figure might help here.

KiloSeq was designed as a method for sequencing arrayed amplicons on a large scale (to more economically meet sequencing needs that are more usually currently met by Sanger capillary sequencing). As such, we would not describe the application of KiloSeq to arrayed libraries as a limitation, but rather a description of the use for which it was designed. We regret that the method was not more clearly described, and have added another Appendix Figure (S10) to illustrate the KiloSeq protocol. Minor note: Although KiloSeq is offered as a service by SeqWell, it was developed by us at the University of Toronto and is not subject to any intellectual property protections and therefore freely available for academic or commercial use.

6. The epistasis analysis in Fig S12 is very interesting but I don't think it was mentioned in the text anywhere. It was surprising to see that most pairwise mutations showing significant epistasis were not near each other. Were any other examples of genetically interacting mutations validated (beyond the I4/P69S example shown) to bolster this finding?

The analysis is described in the Appendix Supplementary text (page 19 of the appendix) and was mentioned, albeit briefly, in the main text on page 6. We agree that the finding that few epistatic pairs corresponded to physically proximal residues was surprising. The I4T-P68S case is one of very few exceptions and it is the only one that we evaluated. It might be interesting to simulate the effects of these variants on folding to see whether more of them could be explained on a biophysical level, but we considered this beyond the scope of this study. We have expanded slightly on the description in the main text, but would be happy to either expand further or remove this topic altogether and save it for a more in-depth future study, at the Editor's discretion.

7. Page 9, "We see three arguments" - four arguments are listed.

Thank you. This has been fixed.

We appreciate the thorough and constructive review process, and would be happy to discuss any of our responses.

2nd Editorial Decision

8 November 2017

Thank you again for sending us your revised manuscript. We have now heard back from the referee who was asked to evaluate your study. As you will see below, the reviewer thinks that the previously raised issues have been satisfactorily addressed. S/he raises however a few remaining concerns, which we would ask you to address in a minor revision.

REVIEWER REPORT

Reviewer #2:

The authors' response addresses most of the concerns in my original review, and I think the revisions yielded a much clearer manuscript. This is a technical tour de force, and will no doubt serve as a reference point for future efforts to systematically measure variant functional impact in human genes.

I do have one small quibble relating to a discrepancy which I only noticed upon reading the revised manuscript. In Table 2, the only two variants called "likely damaging" by DMS were listed as imputed (the scores coming entirely from imputation). However, when I looked up these variants (D94A and D96H), they have appear to have experimentally measured scores, and the final scores shown (joint.score of 0.65 and 0.57, respectively, in CALM1_flipped_scores.csv), do not match those listed in Table 2. The other scores are discrepant as well, but are only off by a small amount (<0.02). With apologies in advance if I am referring to the wrong supplementary table, it'd be great if this could be corrected and any changes in their classification noted.

Reviewer #2:

The authors' response addresses most of the concerns in my original review, and I think the revisions yielded a much clearer manuscript. This is a technical tour de force, and will no doubt serve as a reference point for future efforts to systematically measure variant functional impact in human genes.

We appreciate the kind words.

I do have one small quibble relating to a discrepancy which I only noticed upon reading the revised manuscript. In Table 2, the only two variants called "likely damaging" by DMS were listed as imputed (the scores coming entirely from imputation). However, when I looked up these variants (D94A and D96H), they have appear to have experimentally measured scores, and the final scores shown (joint.score of 0.65 and 0.57, respectively, in CALM1_flipped_scores.csv), do not match those listed in Table 2. The other scores are discrepant as well, but are only off by a small amount (<0.02). With apologies in advance if I am referring to the wrong supplementary table, it'd be great if this could be corrected and any changes in their classification noted.

Good catch. When we performed the blinded comparison with Invitae VUS, we were still working with a previous version of the scoring pipeline. Under the previous pipeline, the two variants in question were classified as poorly measured and therefore were imputed. In the more recent pipeline version, those variants are considered sufficiently well-measured so that they are not imputed. While the data in the supplement and on the website is from our most recent pipeline version, we wished to preserve the integrity of the blinded test so that the scores in Table 2 are those that were in hand at the time we were unblinded to patient indications. Perhaps we were overly conservative here, since the scores from the new pipeline version yielded exactly the same variant classifications and the same P-value measuring association of scores with patient indications. We have now added a note in the manuscript that explains this.

We appreciate the thorough and constructive review process, and would be happy to discuss any of our responses.

YOU MUST COMPLETE ALL CELLS WITH A PINK BACKGROUND

Corresponding Author Name: Frederick P Roth
 Journal Submitted to: Molecular Systems Biology
 Manuscript Number: MSB-17-7908